# Observation of a promethium complex in solution

Darren M. Driscoll[1], Frankie D. White[2], Subhamay Pramanik[1], Jeffrey D. Einkauf[1], Bruce Ravel[3], Dmytro Bykov[4], Santanu Roy[1], Richard T. Mayes[2], Lætitia H. Delmau[2], Samantha K. Cary[2], Thomas Dyke[1], April Miller[2], Matt Silveira[2], Shelley M. VanCleve[2], Sandra M. Davern[2], Santa Jansone-Popova[1], Ilja Popovs[1✉] & Alexander S. Ivanov[1✉]

Lanthanide rare-earth metals are ubiquitous in modern technologies[1–5], but we know little about chemistry of the 61st element, promethium (Pm)[6], a lanthanide that is highly radioactive and inaccessible. Despite its importance[7,8], Pm has been conspicuously absent from the experimental studies of lanthanides, impeding our full comprehension of the so-called lanthanide contraction phenomenon: a fundamental aspect of the periodic table that is quoted in general chemistry textbooks. Here we demonstrate a stable chelation of the $^{147}$Pm radionuclide (half-life of 2.62 years) in aqueous solution by the newly synthesized organic diglycolamide ligand. The resulting homoleptic Pm$^{III}$ complex is studied using synchrotron X-ray absorption spectroscopy and quantum chemical calculations to establish the coordination structure and a bond distance of promethium. These fundamental insights allow a complete structural investigation of a full set of isostructural lanthanide complexes, ultimately capturing the lanthanide contraction in solution solely on the basis of experimental observations. Our results show accelerated shortening of bonds at the beginning of the lanthanide series, which can be correlated to the separation trends shown by diglycolamides[9–11]. The characterization of the radioactive Pm$^{III}$ complex in an aqueous environment deepens our understanding of intra-lanthanide behaviour[12–15] and the chemistry and separation of the $f$-block elements[16].

One reason promethium (Pm) was so elusive for many years, despite a relatively low atomic number, is that it is the only element in the lanthanide (Ln) series (elements with atomic numbers 57–71) with no stable isotopes. Nowadays, mostly synthetic radioisotope $^{147}$Pm (with half-life $\tau_{1/2} = 2.62$ years) is produced and isolated in small quantities through nuclear fission in reactors and subsequent tedious purification steps for many applications. Promethium uses range from long-life nuclear batteries used in space craft to radiation therapy[7,8]. A key obstacle impeding the efficient recovery of this critical element resides in our limited comprehension of the Pm coordination chemistry. In contrast to other lanthanides that favour the +3 oxidation state under ambient conditions, even the most fundamental characteristics of Pm in aqueous solution, including the bond distances and coordination number, remain unexplored. This valuable information is exceptionally challenging to obtain due to its radioactivity, synthetic nature and lack of availability. Only a few simple inorganic Pm$^{III}$ solids, such as halides[17], oxide[18], oxalate[19], molybdate and tungstate[20] have been prepared and characterized by X-ray powder diffraction to determine the lattice parameters. Furthermore, the absorption bands in the visible spectrum[17,21,22], Raman spectra[23] and magnetic susceptibility[24] of the Pm$^{III}$ oxide and halides were reported. Beyond these examples, the fundamental chemistry of Pm is virtually unknown, and there are no

experimental data to benchmark theoretical models for predicting Pm chemical bonding, structure and reactivity in solution. In addition, it is well known that the gradual population of the 4$f$ electron shell in conjunction with relativistic effects cause a continuous decrease in the size of the ionic radii along the lanthanide series, leading to structural changes in Ln complexes. Whereas this lanthanide contraction phenomenon taught in general chemistry textbooks has been inferred mostly from theory[25–29] and Shannon's effective ionic radii database[30], it still lacks experimental structural evidence for a complete set of lanthanides in solution that includes radioactive Pm[31–36]. Advancing our fundamental knowledge in this field is critical for rationalizing and predicting the structurally diverse coordination chemistry shown by lanthanides[1,3,12].

Towards this goal, we report our experimental and computational efforts to investigate the Pm ion binding by a multidentate ligand in an aqueous solution, taking advantage of the recently enhanced isotope separation techniques (Methods), which have enabled the production of $^{147}$Pm in sufficient quantities and purity levels necessary for fundamental studies (Fig. 1a). A new, water-soluble complexing agent, bispyrrolidine diglycolamide (PyDGA) (Fig. 1b) was synthesized and used for Pm complexation. The DGA family of neutral ligands is well established for efficient lanthanide and actinide chelation and

[1]Chemical Sciences Division, Oak Ridge National Laboratory, Oak Ridge, TN, USA. [2]Radioisotope Science and Technology Division, Oak Ridge National Laboratory, Oak Ridge, TN, USA. [3]National Institute of Standards and Technology, Gaithersburg, MD, USA. [4]National Center for Computational Sciences, Oak Ridge National Laboratory, Oak Ridge, TN, USA. ✉e-mail: popovsi@ornl.gov; ivanova@ornl.gov

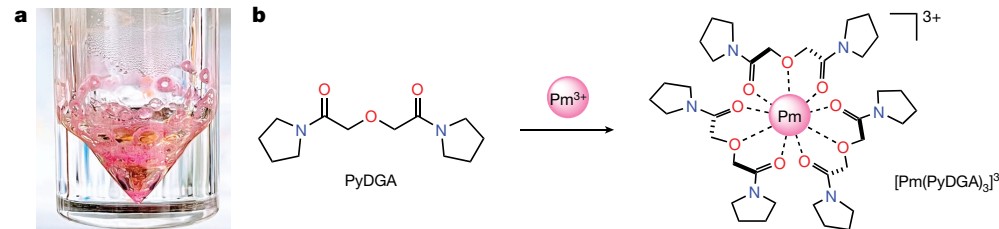

**Fig. 1 | Preparation of PmIII and its chelation by the multidentate ligand PyDGA in an aqueous solution. a**, Photograph of purified PmIII compound prepared in this study. The depicted pink-coloured [147]Pm(NO₃)₃·$n$H₂O ($n < 9$) solid residue was obtained after several purification steps and used in a PmIII complexation. **b**, Each PyDGA ligand molecule consists of two amide carbonyl oxygen groups and one ether oxygen atom, enabling high aqueous solubility. This chelator coordinates with the promethium cation in a tridentate fashion to form the 1:3 complex by providing nine metal-binding O donor atoms in the first coordination sphere of PmIII.

separation[9,10], showing stable binding mode for LnIII ions[37,38]. These characteristics enabled the detection and characterization of the homoleptic [Pm(PyDGA)₃]³⁺ complex by X-ray absorption spectroscopy (XAS) measurements at the National Synchrotron Light Source II (NSLS-II). The experimental results corroborated by the quantum chemical calculations provide the missing piece necessary for a comprehensive study of the impact of $f$-electron count on Ln contraction in the entire isostructural series of Ln complexes. This discovery reveals distinctive structural and electronic characteristics extending beyond the gradual ionic radii changes.

Our motivation for probing the Pm complexation in the solution phase arises from the absence of crystal lattice effects that could affect the measured bond distances. Also, a dilute aqueous environment is generally free from the heat and damage inherent to radioactive materials, which are more pronounced in the solid state. Thus, for XAS investigations, the sample was prepared in 0.01 M HNO₃ solution containing [147]Pm (90 µl, 8.5 mM) complexed with PyDGA at a roughly 1 to 20 metal ion-to-ligand ratio to ensure the full ion chelation and formation of the [Pm(PyDGA)₃]³⁺ complex (Fig. 1b). The solution was then triply contained and secured in a rigid aluminium sample holder (Extended Data Fig. 1). The L₃- and L₁-edge XAS spectra were acquired at room temperature in fluorescence mode using a Vortex four-element silicon-drift detector at beamline 6-BM of NSLS-II (Fig. 2; L₁-edge XAS results are shown in Extended Data Fig. 2). The X-ray absorption near-edge structure (XANES) indicates that the Pm spectral features are consistent with the XANES data measured for other adjacent lanthanides having +3 oxidation state (Extended Data Fig. 3a). The position of the L₃-absorption edge (the inflection point) was determined to be at 6,464.4 eV (calibrated to the K-edge of an Fe foil, 7,112.0 eV, and measured with an instrumental uncertainty below 0.1 eV). The XANES spectrum can be separated into four distinct regions (Fig. 2a). On the basis of our density functional theory (DFT) restricted open shell configuration interaction singles (DFT–ROCIS) and multiple scattering theory calculations (Extended Data Fig. 3b,c), region I corresponds to transitions from Pm 2$p$ to 4$f$/5$d$ orbitals, and the most intense peak II is dominated by 2$p$ core electron excitations to 5$d$ but with some PyDGA orbital contributions. Less visible peak III can be attributed to transitions involving Pm 4$f$/5$d$/ligand orbitals, whereas the origin of broad feature IV is complex with leading components from 2$p$ to 5$d$/ligand and Pm 4$f$ d$z^3$ orbitals.

To investigate the local coordination structure around the PmIII ion, we analysed the extended X-ray absorption fine structure (EXAFS) of the [Pm(PyDGA)₃]³⁺ complex. The Pm EXAFS data in Fig. 2b show the expected sinusoidal-like behaviour; however, a sharp feature can be seen at 8.2 Å⁻¹, which is attributed to the presence of a small amount of NdIII (L₂-edge) and [147]SmIII (L₃-edge), a decay product of [147]Pm. Hence real-space functions (Fig. 2c) were produced using a Fourier transform of the EXAFS data that contain only Pm information ($2.3 \le k \le 7.8$ Å⁻¹), giving a physical description of the atomic arrangement around the Pm ion. The Fourier transform of the EXAFS reveals two intense features at 1.9 and 2.8 Å (non-phase corrected), presumably corresponding to the inner-sphere Pm–O and more distant Pm–C scattering correlations originating from PmIII complexation by PyDGA. A two-shell oxygen-carbon model based on the Pm surrogate crystal structure (Extended Data Fig. 4) was developed to fit the EXAFS data. According to this representation, the first shell comprised six amide carbonyl and three ether O donors, and the second shell at longer distances accounted for the six $sp^3$-hybridized (ether moiety) and six $sp^2$-hybridized (carbonyl moiety) C atoms from the PyDGA scaffold. However, given the limited $k$-space data, affecting the interatomic resolution, and the dynamic nature of the solution phase at room temperature, amido and ether O, as well as $sp^3$- and $sp^2$-C distances, could not be resolved and were each fitted in a conservative manner with the Ln–O and Ln–C single scattering paths. This resulted in an average Pm–O bond distance of 2.476(16) Å (Debye–Waller factor $\sigma^2 = 0.006(1)$ Å²) and an average Pm–C distance of 3.38(7) Å ($\sigma^2 = 0.02(1)$ Å²), consistent with PmIII chelation by three PyDGA ligands (Extended Data Table 1).

To further gain insights into the dynamic structural behaviour of PmIII complexation in an aqueous environment, we performed ab initio molecular dynamics (AIMD) simulations. The theoretical EXAFS spectrum and its Fourier transform (Fig. 2b,c) were simulated directly from the AIMD trajectory and show very good agreement with the experimental data, validating the formation of a homoleptic [Pm(PyDGA)₃]³⁺ complex (Fig. 2d). Key structural parameters align well with those determined by the EXAFS experiments, as can be judged from the analyses of radial distribution functions (RDFs), with the AIMD predicted Pm–O bond length of 2.48 Å (Extended Data Fig. 5). Beyond the inner-sphere Pm–O correlations, the AIMD results also indicated some water structuring around the complex at 4.43 Å through transient hydrogen bond interactions with the O donor groups of the PyDGA ligands. It is also worth noting that, like in the experimental EXAFS data, the amide carbonyl and etheric Pm–O bonds could not be resolved in the AIMD and thus appeared as a single peak in the corresponding RDF, pointing to the dynamic nature of the first-sphere ligand-metal interactions in aqueous solution (Supplementary Video 1).

Next, we performed natural bond orbital (NBO) calculations to examine the nature of Pm–O bonding. Natural population analysis indicates that the promethium 5$d$ and 6$s$ orbitals are substantially populated (0.82 electrons |e| and 0.17 |e|, respectively), with a non-negligible population of the vacant 4$f$ orbitals (0.07 |e|). The dative Pm–O bonds originate from a characteristic σ-type donation of electron density from O lone pairs to the Pm centre. Figure 2e shows the representative leading orbital interaction that stems from an overlap of the O lone pair with an acceptor orbital of primarily 5$d$ character on Pm, resulting in the Pm–O NBO, which is predominantly localized on the oxygen atom (Fig. 2f). The strength of interactions involving amide carbonyl O groups was found to be only slightly higher than that involving ether oxygens. This was confirmed by the comparable calculated values of Wiberg bond

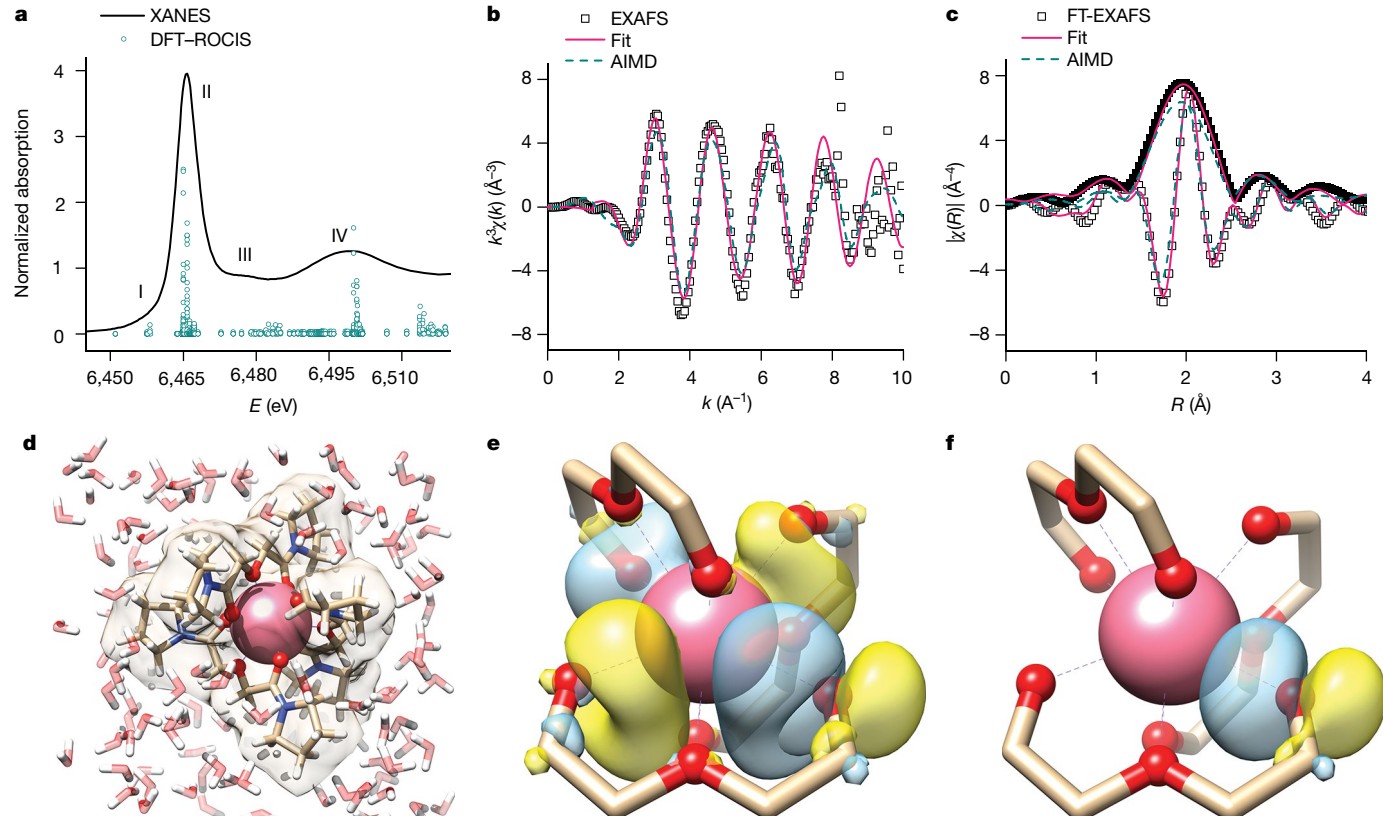

**Fig. 2 | The spectroscopic, structural and electronic characteristics of the observed [Pm(PyDGA)$_3$]$^{3+}$ coordination complex in aqueous environment revealed by synchrotron XAS and quantum chemical studies. a**, Pm L$_3$-edge XANES spectrum (black line) and its interpretation using DFT–ROCIS calculations (circles). $E$ is the incident photon energy and the corresponding orbitals participating in the core electron excitations are shown in Extended Data Fig. 3b. **b,c**, Pm EXAFS data (squares), the fit (pink line) representing model scattering paths associated with the Pm complex and the AIMD simulated EXAFS (turquoise dashed line). **b**, L$_3$-edge EXAFS spectrum of the Pm complex in solution where $k$ is the energy of the photoelectron in wavenumbers and $k^3\chi(k)$ is the $k^3$-weighted EXAFS function. Data between 2.3 and 7.8 Å$^{-1}$ were

Fourier transformed using a Hanning window to obtain real-space information. **c**, Magnitude of the Fourier transform (FT) (black squares) and the real component of the Fourier transform (empty squares). The data were fit over the range from 1.4 to 3.2 Å. Spectra are not phase adjusted. **d**, Snapshot of the Pm complex surrounded by water molecules from the AIMD simulations. **e**, Formation of the dative Pm–O bond in the Pm complex in terms of overlapping amide carbonyl oxygen lone pair, on the right, with the Pm 5$d$ acceptor orbital, on the left. Only the local Pm–ligand environment is visualized for clarity. **f**, The resulting Pm–O bonding NBO that includes roughly 4% Pm character. The Pm hybrid's nodal character in the bond is not visible because its amplitude is below the 0.035 amplitude cut-off for the orbital visualization.

indices for the amidic (0.12) and etheric (0.08) Pm–O bonds, pointing to their prevalent ionic nature and explaining their dynamic behaviour in aqueous solution. As a result, these bonding characteristics do not exert substantial ligand field effects, leading to the challenges that are frequently encountered in the selective recovery of Pm and other rare-earth elements[1].

Having established promethium coordination and bond lengths, we studied the remaining lanthanide (La$^{III}$, Ce$^{III}$, Pr$^{III}$, Nd$^{III}$, Sm$^{III}$, Eu$^{III}$, Gd$^{III}$, Tb$^{III}$, Dy$^{III}$, Ho$^{III}$, Er$^{III}$, Tm$^{III}$, Yb$^{III}$, Lu$^{III}$) complexes with PyDGA using XAS (Fig. 3 and Extended Data Fig. 6) to understand how the solution structure of the coordination complex transforms across the lanthanide series. The Fourier transform-EXAFS results in Fig. 3a,b show that the positions and intensities of the main features corresponding to the Ln–O distances vary slightly between the lanthanides. This is expected on the basis of the different harmonics generated from the shortening of the inner-sphere bonds caused by the Ln contraction. Furthermore, the shrinkage of the Ln–O bonds is corroborated by the trend in the relative energy positions of the Ln L$_3$-edge XANES spectral features (Extended Data Fig. 3a), consistent with the results of a recent study[39] on some isostructural Ln compounds using high-energy-resolution fluorescence-detected XANES[40,41] measurements.

Good fits to the Ln-PyDGA EXAFS data were obtained with the model used for promethium, giving physically sound parameters (Extended

Data Table 2) and suggesting that the [Ln(PyDGA)$_3$]$^{3+}$ species prevail in the aqueous solution across the series. Figure 3c represents the most comprehensive view of the Ln contraction phenomenon obtained from experiments and shows how the inner-sphere Ln–O bond distances change depending on the number of 4$f$ electrons in the electronic structure of Ln$^{III}$. By monitoring the decrease of Ln–O bonds from La (2.560(21) Å) to Lu (2.329(12) Å), a quadratic dependence across the series was observed and fitted by a polynomial regression (Extended Data Fig. 7). Filling the 4$f$ orbitals apparently influences shielding of the nuclear charge and according to our data this effect was most pronounced early in the series from La to Pm, accounting for as much as roughly 36% of the overall Ln contraction. After Pm, there was a steadier shortening of the Ln–O bonds. This behaviour is in line with Shannon's effective ionic radii decrease (at coordination number of nine)[30], which is larger at the beginning of the series than at the end. It is also worth mentioning that the observed accelerated contraction parallels well with the Ln extraction performance of lipophilic diglycolamides in a liquid–liquid extraction process, where better separation between adjacent lanthanides was achieved for the light (La–Nd) than for the heavy (Er–Lu) members of the series[9,10]. Moreover, by adapting the modified Slater theoretical model[36] to our experimental dataset, we derived a value of the shielding constant for $f$ electrons ($s = 0.74$), which is in good agreement with the previously reported and generally accepted

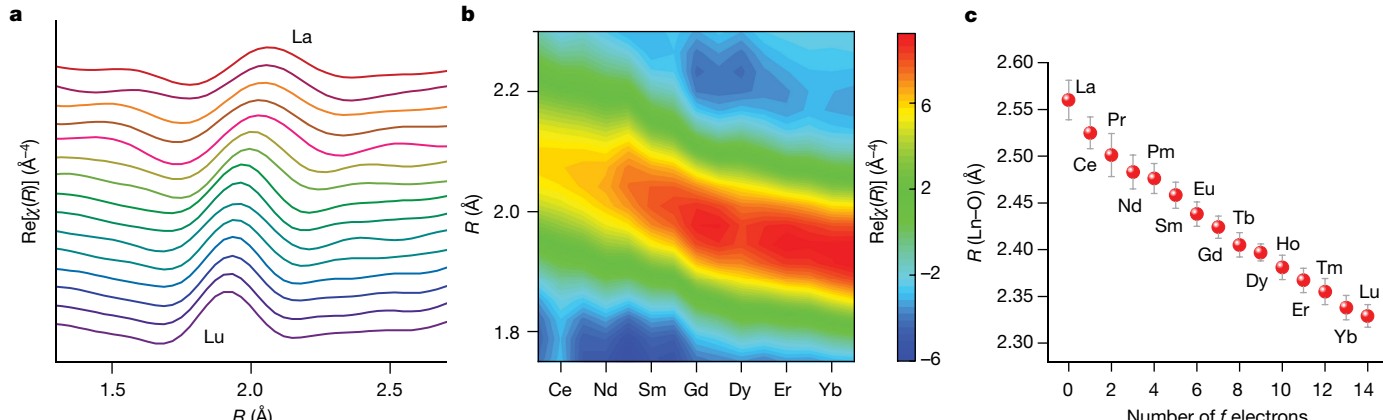

**Fig. 3 | The lanthanide contraction phenomenon captured by the element-specific XAS for the entire isostructural series of the lanthanide complexes in solution. a,b,** One-dimensional profiles (**a**) and 2D intensity map (**b**) of the real component of the Fourier transformed EXAFS data for the lanthanide complexes, visualizing the contraction of the first shell across the lanthanide series. Spectra are not phase adjusted. **c,** The dependence of the Ln–O bond distances on the number of 4*f* electrons, revealing accelerated contraction from La$^{III}$ to Pm$^{III}$ followed by a steadier Ln–O bond shortening for the heavier lanthanides (1$\sigma$ error bars associated with each data point are based on EXAFS fitting uncertainty).

value of 0.69 obtained from the Ln ionization energies[42]. We note, however, that accurate fully relativistic quantum mechanical calculations using a new generation of supercomputers will be important to further investigate the observed Ln contraction behaviour in future studies.

After almost eight decades since the discovery of the element Pm, its coordination complex has been synthesized and characterized in solution using modern synchrotron spectroscopy tools. The determined Pm–O bond distance of 2.476(16) Å is in line with quantum chemical investigations and originates from a σ-type donation of electron density from the ligands to the primarily 5*d* vacant orbitals of Pm. Finally, this previously inaccessible piece of information allowed us to complete structural studies of a full lanthanide set of isostructural complexes in solution, ultimately establishing and confirming the Ln contraction phenomenon solely based on the experimental structural data. These results are expected to contribute to our fundamental understanding and prediction of the coordination chemistry of lanthanides and scarce *f*-block elements[43–48], with pertinence to emergent rare-earth separation and radiopharmaceutical technologies.

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

## Methods

### Materials synthesis

The PyDGA ligand was synthesized according to the following procedure. In a round-bottom flask equipped with a stir bar, pyrrolidine (11.68 ml, 2.5 equiv.) was combined with anhydrous $CH_2Cl_2$ (120 ml) and $Et_3N$ (19.58 ml, 2.5 equiv.). The reaction mixture was stirred for 15 min in an ice-water bath. Diglycolyl chloride (6.67 ml, 1.0 equiv.) was added dropwise under an inert atmosphere (Argon), and the reaction mixture was allowed to warm up to room temperature, followed by stirring for the next 12 h. Afterwards, $CH_2Cl_2$ was evaporated to dryness under reduced pressure, and the residue was dissolved in 100 ml of methanol and treated with $K_2CO_3$ (23.32 g, 3.0 equiv.) to convert $Et_3N \cdot HCl$ to KCl and free triethylamine. The reaction mixture was filtrated through a short Celite plug and rinsed with excess methanol to separate the solid salt, and the filtrate was concentrated to yield a crude product. The crude product was purified on CombiFlash $R_f$ automated flash chromatography system using normal phase silica gel as a stationary phase and gradient 0–20% MeOH in $CH_2Cl_2$ as an eluent to yield a white crystalline solid (12.00 g, 89%) (see Extended Data Fig. 8 for spectra from [1]H nuclear magenetic resonance, [13]C nuclear magenetic resonance, Fourier transform-infrared spectroscopy and electrospray ionization with mass spectrometry).

### [147]Pm experimental preparation

Caution! [147]Pm ($\tau_{1/2}$ = 2.62 years) has potential health risks due to its β emission. Processing, preparation and handling were carefully performed in a radiological facility with gloveboxes and fume hoods equipped with HEPA (high-efficiency particulate absorbing) filters. The preparation of samples was carefully surveyed and monitored for contamination by trained radiological control technicians.

The promethium was harvested from the waste solutions generated by the production of [238]Pu from irradiated [237]Np targets. The concentration and initial crude separation of promethium was done using a separation column[49] in the hot cell. This had the advantage of obtaining the Pm in a manageable volume and rejecting most other fission products. A careful gradient separation substantially decreased the amount of the high gamma-emitting lanthanide fission products, specifically [141,144]Ce and [154,155,156]Eu. The solution was further purified by repeating separation cycles in smaller columns within shielded caves or gloveboxes, leaving essentially a Pm solution still containing traces of curium as these two elements typically co-extract and costrip during this process. The separation between promethium and curium was accomplished using a TALSPEAK-based solvent extraction system[50] with several scrubs to reach the desired purity.

A 70 mM [147]Pm[III] stock solution in 0.01 M $HNO_3$ was prepared for distribution into the XAS sample after the dilution. To ensure the complete complexation of promethium, a solution (roughly 90 µl) of 8.5 mM [147]Pm$(NO_3)_3$ containing 180 mM PyDGA was prepared. The obtained solution was then loaded into a polyimide capillary (1.8 mm inner diameter by 5 cm long, 0.05 mm thickness, Cole-Parmer) using a Hamilton syringe and then sealed twice with Devcon 2 Ton epoxy (Extended Data Fig. 1). Once the epoxy had dried completely, the sample was transferred from a glovebox to a radiological fume hood for further decontamination. The sample was then surveyed and doubly contained for shipment to the XAS beamline.

The radiochemical purity of the recovered [147]Pm$(NO_3)_3$ used for the [Pm(PyDGA)₃]³⁺ sample preparation was more than 99.9%. The residual concentration of [151]Sm was assessed at below the detection level. A small quantity of [146]Nd was present in the sample due to challenging separations of the adjacent lanthanides using the aforementioned techniques. Traces of Sm present in the sample on the moment of XAS measurements originated from the radioactive decay of promethium to the daughter samarium according to the following process: [147]Pm (β⁻→) [147]Sm. Roughly 77 days had passed between the Pm purification and XAS

data collection. On the basis of the $\tau_{1/2}$ = 2.62 years of the radioactive decay, up to 5.632% of the starting [147]Pm had decayed into [147]Sm at the time of the sample measurements at NSLS-II.

### XAS data collection and analysis

XAS measurements were acquired at the Ln $L_3$- and $L_1$-edges at beamline 6-BM of the NSLS-II. For the dilute solution of Pm, measurements were performed in fluorescence mode using a four-element silicon-drift detector with no beam-induced changes to the sample being detected. This was checked by comparing individual XAS scans, which did not show any abnormal changes. For all other Ln (La–Nd, Sm–Lu), aqueous solutions of 0.1 M Ln$(NO_3)_3$ (prepared from commercial solid Ln[III] nitrate salts with 99.9% metal purity) and 0.4 M PyDGA were combined in 0.01 M $HNO_3$, and then placed into polyether ether ketone (PEEK) liquid holders of varying thickness with polyimide windows and sealed with epoxy, affording XAS data collection in transmission mode. The data for Ln, except Pm, were energy-calibrated to the main edge from the spectra of Ln oxide standards. The Ln dataset consisted of three scans, which were averaged and background subtracted. For the Pm, 20 ($L_3$-edge) and 30 ($L_1$-edge) individual scans were merged with first derivative maxima at 6,464.4 and 7,441.4 eV, respectively (calibrated to the K-edge of an Fe foil, 7,112.0 eV). Data normalization was performed using the Athena software package[51].

Ejected photoelectrons are defined by their wavenumber ($k$) in relation to the absorption edge energy ($E_0$) through the equation

$$k = \sqrt{2m_e(E - E_0)/\hbar^2} \tag{1}$$

where $m_e$ is the electron mass and $\hbar$ is the reduced Planck's constant. The experimental EXAFS oscillations of each sample, $\chi(k)$, were extracted from the normalized XAS data using subtraction of a spline and a cut-off distance ($R_{BKG}$) that varied between 1.2 and 1.0 Å. For analysis of the EXAFS region, we used the EXAFS relationship given by

$$\chi(k) = \sum_i \frac{F_i(k)S_0^2 N_i}{kR_i^2} e^{-2k^2\sigma_i^2} e^{\frac{-2R_i}{\lambda(k)}} \sin\left(2kR_i + \delta_i(k) - \frac{4}{3}k^3 C_{3,i}\right) \tag{2}$$

where the index $i$ is considered the path index and the $\chi(k)$ is calculated as the summation over all paths. For fitting of the EXAFS, FEFF6 within the Artemis software package[51] was used considering the experimental $\chi(k)$ data weighted by $k^3$. In equation (2), $F_i(k)$, $\delta_i(k)$ and $\lambda(k)$ represent the effective scattering amplitude, total phase shift and mean-free-path of the photoelectron and each are derived from the FEFF6 code. The many-body amplitude-reduction factor, $S_0^2$, was fixed to 1. Furthermore, $N_i$ values, the degeneracy of the path and therefore the coordination numbers for single scattering paths, were held constant (9 and 12 for the first and second coordination shells, respectively), as inferred from the stable 1:3 complexation and the respective Ln-DGA crystal structures[37]. Therefore, the parameters still to be fit included $R_i$, the half-path length; $\sigma_i^2$, the Debye–Waller factor; and $C_{3,i}$, the asymmetry of the distribution. Variation of $C_{3,i}$ was found to provide negligible improvements on the single scattering paths and thus was not included in the fitting process. Furthermore, a single non-structural parameter for all paths, $\Delta E_0$, was varied to align the $k = 0$ point of the experimental data and theory. Fits were performed in $R$ space using a Hanning window for $k$-space data. For the EXAFS fits, we focused on the Ln–O and Ln–C single scattering paths originating from the binding of three PyDGA ligands. For all lanthanides, both the $L_3$- and $L_1$-edge spectra were simultaneously fit with only the addition of a second $\Delta E_0$ variable for the $L_1$-edge data. The $L_3$-edge dataset included both the single scattering paths for Ln–O and Ln–C, whereas the $L_1$-edge used a restricted $R$-window and only the Ln–O scattering path was fitted. This approach allowed the number of variables (six) per fit to stay below the number of independent data points (ten) available in the primary Pm data with $k_{max}$ = 7.8 Å⁻¹.

## X-ray diffraction studies

Crystallization of $[Sm(PyDGA)_3][Sm(NO_3)_6] \cdot 3C_2H_5OH$: a solution (1.0 ml) of $Sm(NO_3)_3$ (56 mg, 125 mM) was added to 1 ml of $CH_3OH$:$C_2H_5OH$ (1:1) solution of PyDGA (60 mg, 250 mM), followed by vapour diffusion under isopropyl ether inside a refrigerator at 5 °C. After 7 days, plate shape (monoclinic, $P2_1/c$) crystals were obtained. Crystallization of $[Er(PyDGA)_3]_2[Er(NO_3)_5]_3 \cdot 2H_2O$: a solution (1.0 ml) of $Er(NO_3)_3$ (110 mg, 250 mM) was added to a 1 ml of $CH_3OH$:$H_2O$ (9:1) solution of PyDGA (120 mg, 500 mM), followed by vapour diffusion under diethyl ether at room temperature. After 3 days, triclinic ($P$–1) crystals were obtained. X-ray diffraction data were collected at 100 K on a Bruker D8 Advance Quest diffractometer equipped with a graphite monochromator using Mo $K\alpha$ radiation ($\lambda = 0.71073$ Å). The frames were integrated with the Bruker SAINT software package using a narrow-frame algorithm. An empirical absorption correction using the multi-Scan method SADABS was applied to the data. The structure was solved by direct methods using the Bruker SHELXTL Software Package, v.2018/3. Non-hydrogen atoms were refined anisotropically. Hydrogen atoms were calculated and placed in idealized positions. The CIF files within this report were archived in the Cambridge Crystallographic Data Centre (CCDC) under CCDC depositions 2279633 and 2279634.

## Computational details

The Vienna ab initio simulation package (VASP)[52,53] was used to conduct AIMD simulations using spin-polarized DFT. The valence electronic states were expanded on a basis of plane waves, whereas the core valence interactions were described using the projector augmented wave approach and standard $f$-in-valence projector augmented wave potential was used for Pm[54,55]. The plane-wave kinetic energy cut-off was set to 650 eV and the Perdew–Burke–Ernzerhof (PBE) GGA functional[56] was used to describe the exchange-correlation interactions. The Brillouin zone was sampled using the gamma point approximation. The DFT-D3 approach of Grimme[57] was used to account for the van der Waals interactions. The initial structure of the Pm complex–water system (a periodic cubic box of 18 Å length containing one complex and 144 water molecules) was pre-equilibrated for 5 ns in a canonical ensemble at a temperature of 300 K using the extended polymer consistent force field (PCFF+)[58] supported in MedeA-LAMMPS[59,60]. As the nitrate counterions are expected to be completely dissociated and/or screened from $[Pm(PyDGA)_3]^{3+}$ in a dilute aqueous environment[38], they were not explicitly introduced in the molecular dynamics simulations and the +3 charge on the Pm complex was instead compensated by a uniform background charge. AIMD simulations at 300 K were performed using the Nosé–Hoover thermostat[61,62] with a time step of 1 femtosecond (fs). After equilibrating for 10 picoseconds (ps), the AIMD trajectory was collected for 50 ps and used for the RDF analysis. Furthermore, the evenly spaced 1,000 configurations from the last 10 ps of the AIMD trajectory were used to compute and simulate AIMD-EXAFS spectra using the Green's function-based approach implemented in the FEFF9 package[63]. Before running the FEFF9 code, a coordinate transformation procedure was performed to ensure that the absorbing ion, Pm, was at the centre of the simulation box and the other atoms were arranged according to their distances from Pm in the ascending order. The multiple scattering path expansion within 8.5 Å of Pm was used during the self-consistent cycle. All multiple scattering paths were included within the plane-wave approximation except the ones with the mean amplitude below 0.01%. XANES and projected density of states calculations of the Pm complex were also performed using FEFF9 (ref. 63). The XANES spectrum was computed using the full multiple scattering, self-consistent field and Hedin–Lundqvist energy-dependent exchange-correlation potential, considering both dipolar and quadrupolar transitions. The ground state potential was used for the background function. For the projected density of states calculations, a Lorentzian broadening parameter of 0.05 eV was applied.

Cluster model calculations in the gas phase were performed with the Gaussian v.16, Revision A.03 program package[64]. Geometry optimizations enlisted unrestricted Kohn–Sham methods, with the aug-cc-pVTZ basis set for the light atoms[65]. The small-core $f$-in-valence quasi-relativistic ECP28MWB/ECP28MWB_ANO effective-core-potential/basis-set[66] was used for Pm and the complex was treated as a triply charged quintet with four unpaired $f$ electrons. The optimized structure at the PBE0-D3 level of theory[67] was confirmed as a true minimum by analytical frequency calculations. The Pm first- and second-sphere bond distances agreed well with the EXAFS (Extended Data Table 1) and AIMD data (Extended Data Fig. 5), and this structure was used for our subsequent analysis. The Pm $L_3$-edge XANES calculations were performed with the ORCA v.5.0 program[68]. The ROCIS method was used on top of the DFT wave function (DFT–ROCIS)[69,70]. The B3LYP functional[71] was deployed together with Douglas–Kroll–Hess (DKH) Hamiltonian to account for relativistic effects. The DKH-optimized all-electron TZ-quality basis set was applied to all elements except for Pm, in which segmented all-electron relativistically contracted basis was used (the dkh-def2-tzvp and sarc-dkh-tzvp in ORCA notation, correspondingly). Spin–orbit coupling as well as lower and higher multiplets were accounted for. The analysis was done using the natural difference orbitals[72]. To account for systematic errors in the calculation of transition energies, the simulated spectrum was uniformly shifted by 175 eV to match the experimental absorption edge energy. The bonding in the Pm complex was examined by using the NBO methodology[73], as implemented in the NBO7 program[74,75]. Molecular orbital diagrams were drawn with an isovalue of 0.035 a.u. Model representations in the figures were prepared using the UCSF Chimera software[76]. The Slater shielding constant for $4f$ electrons was derived based on the methodology described by Seitz et al.[36].

## Data availability

All data supporting the findings are available within the paper. Additional details are available on request to the corresponding authors. The X-ray crystallographic data for the Sm and Er-PyDGA structures reported in this study have been deposited at the CCDC, under deposition numbers 2279633 and 2279634, respectively. These data can be obtained free of charge from the CCDC via www.ccdc.cam.ac.uk/data_request/cif.

## Code availability

All in-house code used in this study is available via Zenodo at https://doi.org/10.5281/zenodo.10045182 (ref. 77).

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

**Acknowledgements** This research was supported by the US Department of Energy (DOE), Office of Science, Office of Basic Energy Sciences, Chemical Sciences, Geosciences, and Biosciences Division and Materials Sciences and Engineering Division under award number DE-SC00ERKCG21 (D.M.D., S.P., S.R., S.J.-P. and A.S.I.); the DOE Isotope Programme, managed by the Office of Science for Isotope R&D and Production (F.D.W., R.T.M., L.H.D., S.K.C., T.D., A.M., M.S., S.M.V., S.M.D. and I.P.); and the DOE, Office of Science, Office of Basic Energy Sciences, Chemical Sciences, Geosciences, and Biosciences Division under award number DE-SC00 ERKCC08 (J.D.E.). Use of the NSLS-II (NIST beamline 6-BM) was supported by the DOE Office of Science User Facility operated for the DOE Office of Science by Brookhaven National Laboratory under contract no. DE-SC0012704. This research used resources of the Oak Ridge Leadership Computing Facility (OLCF) and the Compute and Data Environment for Science (CADES) at the Oak Ridge National Laboratory, which is supported by the Office of Science of the DOE under contract no. DE-AC05-00OR22725. This research used the hot cells and glovebox laboratories and other resources of the Radiochemical Engineering Development Centre, a DOE Office of Science research facility operated by the Oak Ridge National Laboratory. D.M.D., B.R., I.P. and A.S.I. thank K. Wehunt of Brookhaven National Laboratory for her help with handling radioactive samples at NSLS-II and E. Jahrman of the National Institute of Standards and Technology for critically reading the manuscript and providing helpful suggestions. I.P. and A.S.I. thank R. Copping, L. Harvey, N. Sims and M. Du for helpful discussions.

**Author contributions** I.P., A.S.I. and S.J.-P. acquired funding. I.P. and A.S.I. conceived and led the project, conceptualized the study and wrote the first draft. S.P., S.J.-P. and I.P. synthesized the ligand. F.D.W., R.T.M., L.H.D., S.K.C., T.D., A.M., M.S., S.M.V. and S.M.D. produced, purified and prepared the Pm XAS sample. S.P. and D.M.D. prepared non-radioactive lanthanide samples. D.M.D. and A.S.I. acquired XAS beamtime at NSLS-II. D.M.D., B.R. and A.S.I. designed and conducted XAS experiments. D.M.D. analysed, fitted and summarized XAS data. S.P. obtained single crystals of Ln-PyDGA complexes. J.D.E. collected and refined single-crystal X-ray diffraction crystallographic data. A.S.I. and D.B. acquired computational time on the OLCF. A.S.I. performed and interpreted AIMD simulations and chemical bonding analysis. S.R. simulated AIMD-EXAFS and XANES spectra using FEFF9. D.B. performed and interpreted XANES calculations using DFT–ROCIS. All authors discussed the results and contributed to the final manuscript.

**Competing interests** The authors declare no competing interests.

**Additional information**
**Correspondence and requests for materials** should be addressed to Ilja Popovs or Alexander S. Ivanov.

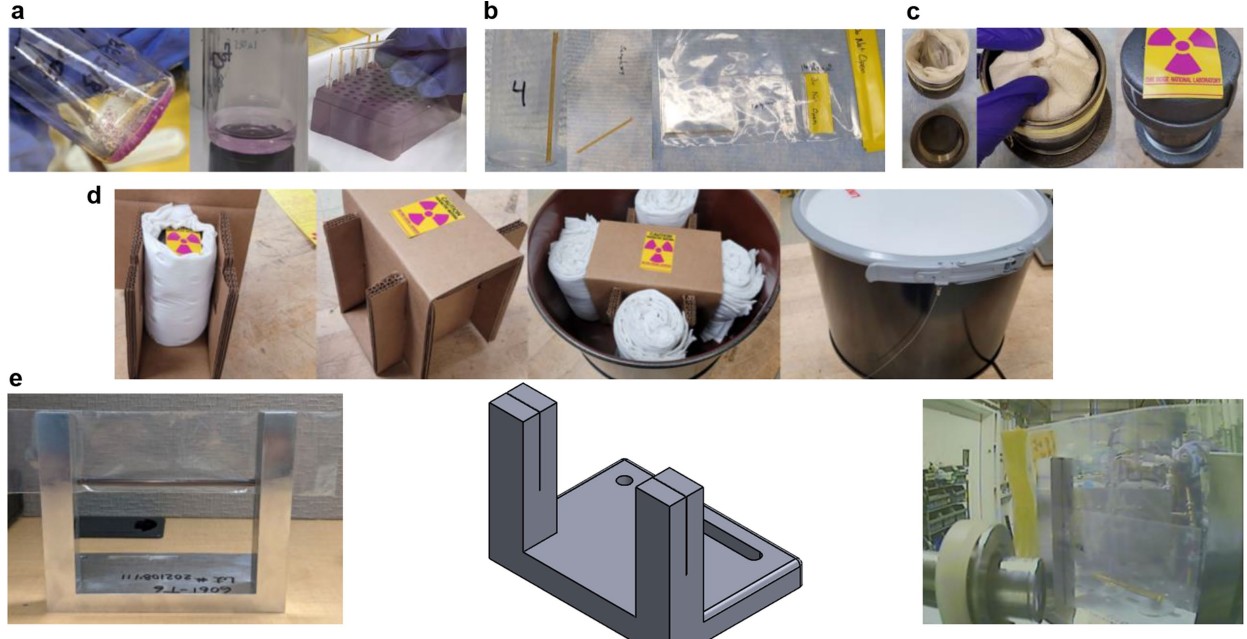

**Extended Data Fig. 1 | Pm sample preparation and transportation steps for X-ray absorption spectroscopy measurements. a**, (left) $^{147}$Pm(NO$_3$)$_3 \cdot n$H$_2$O ($n < 9$) solid residue; (middle) 70 mM 0.01 HNO$_3$ $^{147}$Pm(NO$_3$)$_3$ stock solution; (right) $^{147}$Pm-PyDGA sample being epoxied before removal from glovebox. **b**, (left) fully sealed Kapton capillary with solution of $^{147}$Pm-PyDGA; (middle) capillary sealed within one polypropylene bag; (right) capillary sealed within two polypropylene bags. **c**, Shipping preparations: (left) folded triple bagged Kapton capillary with solution of $^{147}$Pm-PyDGA inside pipe nipple along with absorbent material; (middle) folding in of absorbent material for cap placement; (right) cap hand tightened, and then radiological label applied. **d**, Shipping preparations: (left) pipe nipple wrapped in absorbent material and then placed inside of cardboard insert; (middle) cardboard insert put inside of 5-gal drum along with absorbent packaging material; (right) wire looped through drum ring. **e**, (left) demonstration of empty capillary within polypropylene bag held by the aluminium sample holder designed for the $^{147}$Pm-PyDGA measurements; (middle) 3D drawing of the sample holder used for the $^{147}$Pm-PyDGA measurements; (right) the Pm sample photograph from the beamline camera taken during XAS measurements.

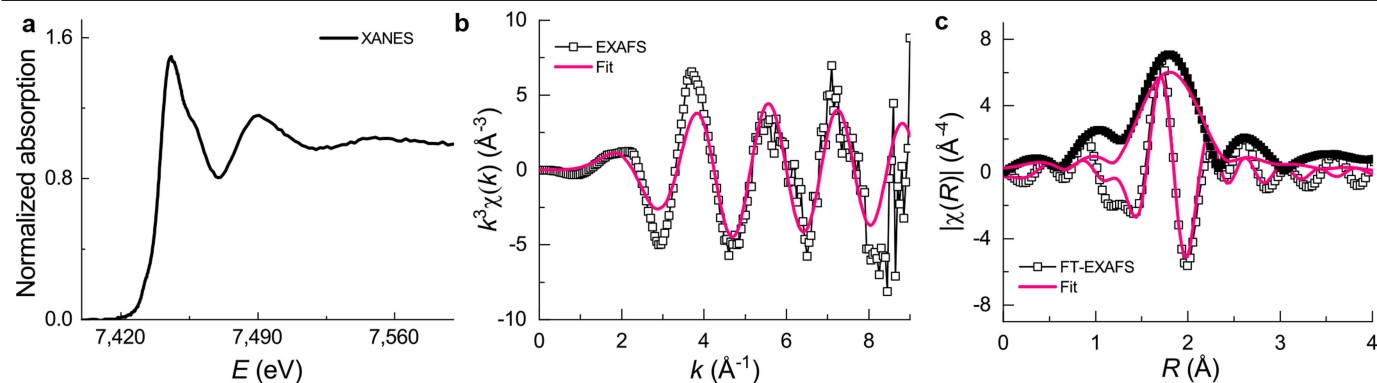

**Extended Data Fig. 2 | L$_1$-edge XAS data for the [Pm(PyDGA)$_3$]$^{3+}$ complex in solution at room temperature. a**, Pm L$_1$-edge XANES spectrum (black line). **b-c**, Pm EXAFS data (squares) and the fit (pink line). (**b**) L$_1$-edge EXAFS spectrum of the Pm complex where $k$ is the energy of the photoelectron in wavenumbers and $k^3\chi(k)$ is the $k^3$-weighted EXAFS function. (**c**) Magnitude of the Fourier transform (black squares) and the real component of the Fourier transformed EXAFS data (empty squares).

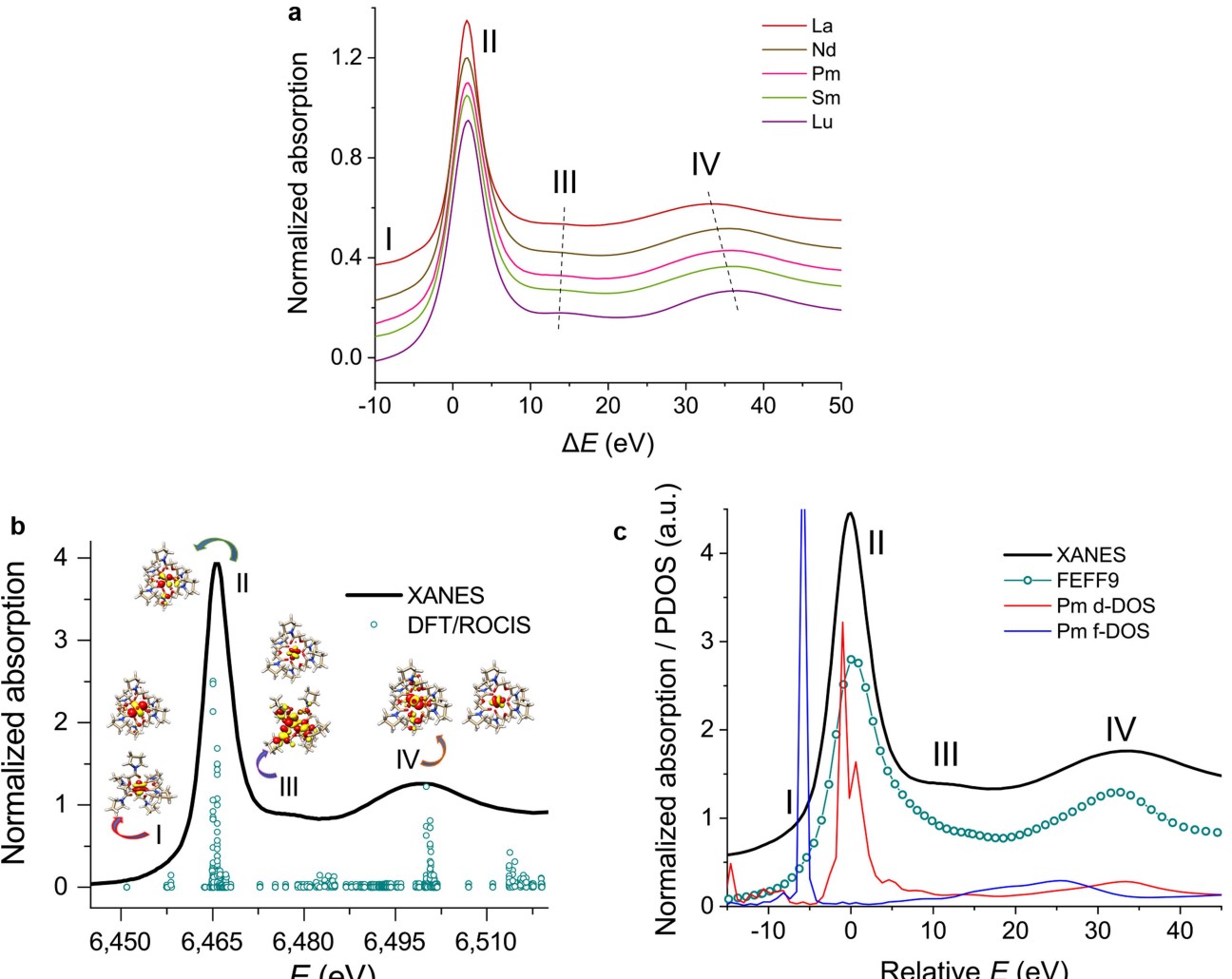

**Extended Data Fig. 3 | Comparison of L$_3$-edge XANES data for the selected lanthanide [Ln(PyDGA)$_3$]$^{3+}$ complexes in solution and Pm L$_3$-edge XANES spectrum and its interpretation using DFT/ROCIS and multiple scattering (FEFF9) calculations. a**, Nd$^{III}$ and Sm$^{III}$ spectra are compared to the Pm$^{III}$ data, confirming the +3 oxidation state. The energy separation between the white line (II) and the first postedge feature (III) decreases, whereas the energy separation between the white line (II) and the second postedge peak (IV) increases across the Ln series. The obtained trend is consistent with a previous study[39] using HERFD-XANES, where the shift to higher energies of peak IV was attributed to lanthanide contraction (shortening of the inner-sphere bonds across the Ln series). The plot is presented as a function of $\Delta E$ (the difference between the photon energy $E$ and the peak in the first derivative of the data $E_0$).

The spectra are scaled to the same maximum height and offset for clarity. Dashed lines are guides to the eye. **b**, Experimental (black line) and simulated XANES spectra using DFT/ROCIS calculations (circles) with the representative orbitals participating in the core electron excitations, which correspond to different regions of the XANES spectrum. Band assignment was performed based on natural difference orbitals (NDOs), drawn with 0.03 au isosurface value. Only the acceptor NDOs are visualized. **c**, Comparison of experimental (black line) and simulated XANES spectra using FEFF9 calculations (circles) with the projected density of states (PDOS) related to the Pm$^{III}$ $d$ and $f$ orbital contributions. To compare the results on a common energy scale, the maximum of the absorption edge has been set to zero. The spectra are offset for clarity.

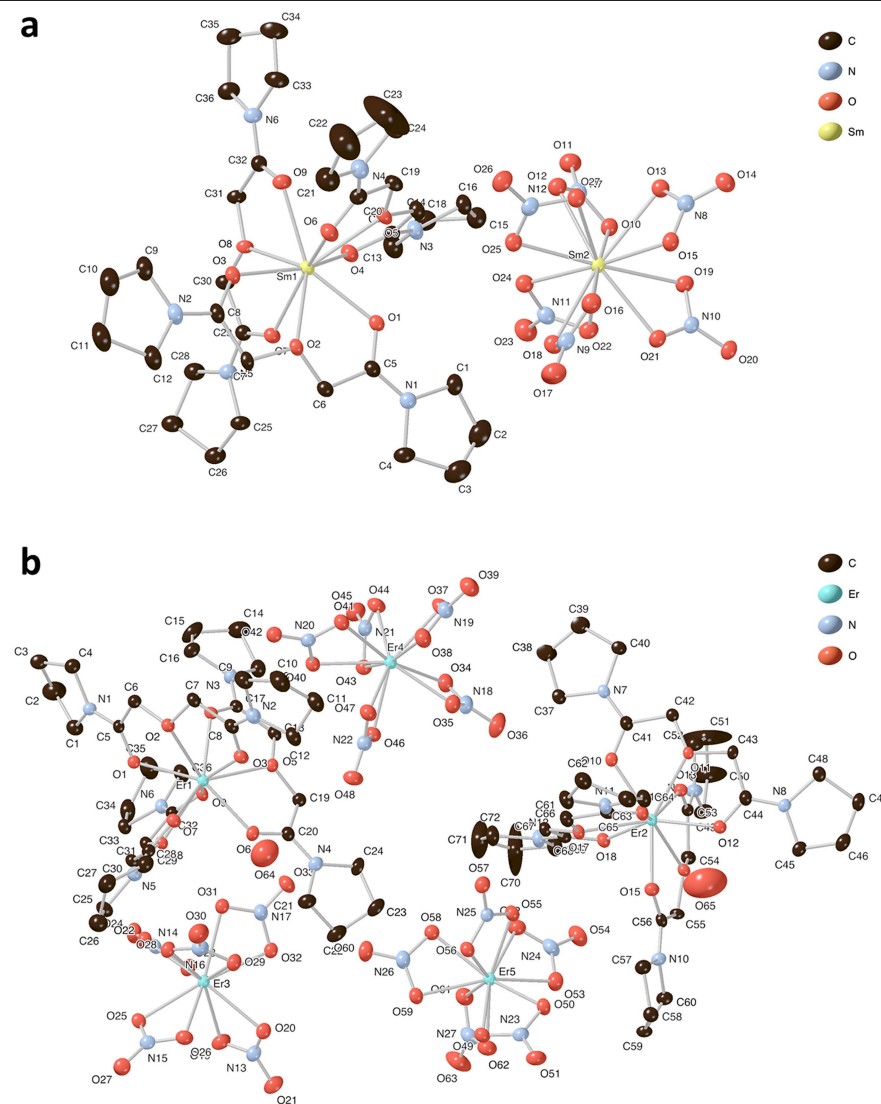

**Extended Data Fig. 4 | X-ray crystal structures of the Pm surrogate [Sm(PyDGA)₃][Sm(NO₃)₆]·3C₂H₅OH and [Er(PyDGA)₃]₂[Er(NO₃)₅]₃·2H₂O complexes. a**, Thermal ellipsoid plot (50% probability level) of [Sm(PyDGA)₃] [Sm(NO₃)₆]·3C₂H₅OH crystals (CCDC:2279633). Hydrogen atoms and solvents are omitted for clarity. **b**, Thermal ellipsoid plot (50% probability level) of [Er(PyDGA)₃]₂[Er(NO₃)₅]₃·2H₂O crystals (CCDC: 2279634). Hydrogen atoms are omitted for clarity.

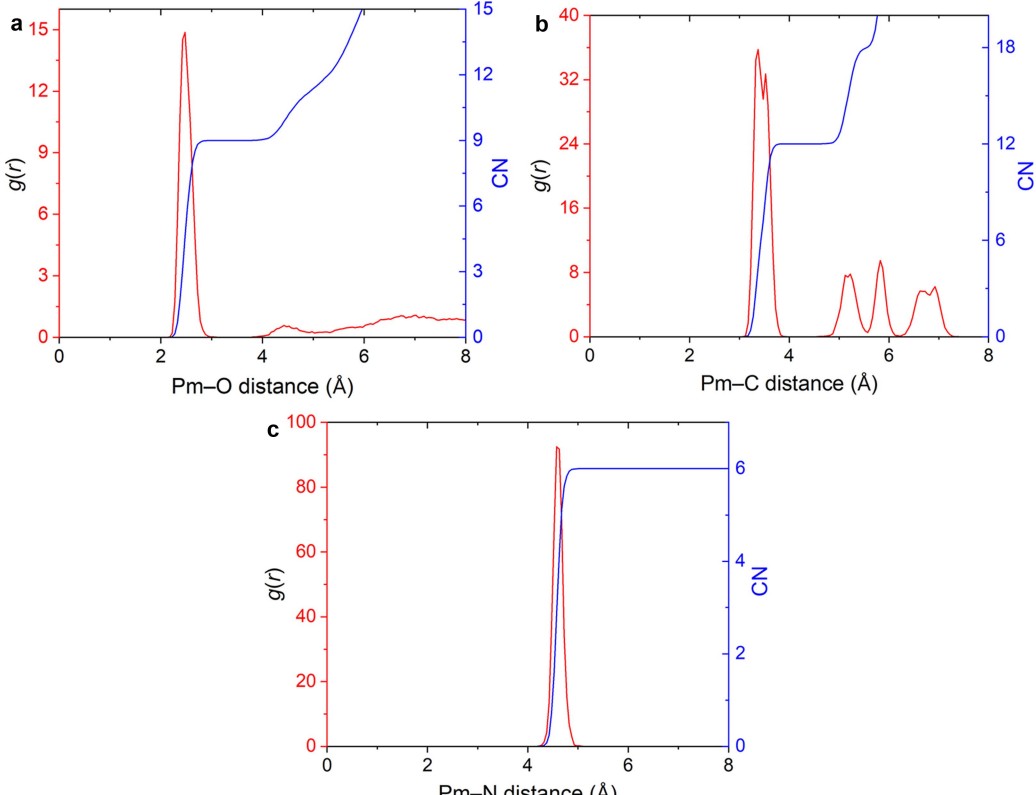

**Extended Data Fig. 5 | Structural parameters for the [Pm(PyDGA)₃]³⁺ complex in aqueous solution obtained from AIMD simulations.** Radial distribution function ($g(r)$; red curve, left axis) and its integration (coordination number, CN; blue curve, right axis) of (**a**) oxygen atoms, including PyDGA donor atoms and water molecules, (**b**) PyDGA carbon atoms, and (**c**) PyDGA nitrogen atoms around Pm^III. Water structuring around the complex at 4.43 Å can be observed due to transient hydrogen bond interactions with the O donor groups of PyDGA ligands. As can be seen, the amide carbonyl and etheric Pm–O bonds could not be resolved at room temperature due to their dynamic nature in solution. However, the simulations show distinct Pm–C correlations with the peaks corresponding to the sp³- and sp²-C positions relative to Pm^III, pointing to their more rigid behavior upon complexation. The AIMD average bond lengths (Pm–O distance of 2.48 Å and Pm–C distance of 3.44 Å) agree well with the results of static DFT calculations (Pm–O distance of 2.47 Å and Pm–C distance of 3.44 Å) and the EXAFS data in Extended Data Table 1.

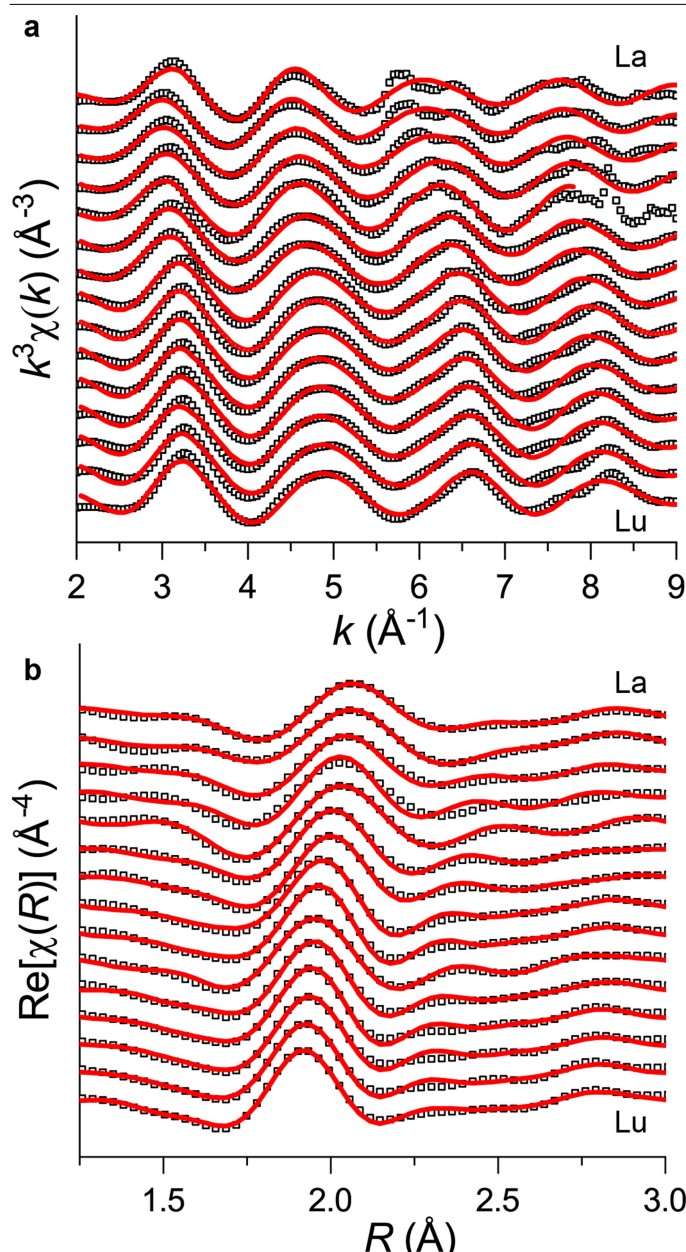

**Extended Data Fig. 6 | EXAFS data (squares) and the fit (red line) for the entire set of isostructural [Ln(PyDGA)$_3$]$^{3+}$ complexes in solution. a**, L$_3$-edge EXAFS spectra of the lanthanide complexes in solution where $k$ is the energy of the photoelectron in wavenumbers and $k^3\chi(k)$ is the $k^3$-weighted EXAFS function. The apparent features in the experimental EXAFS data at approximately 5.7 Å$^{-1}$ to 6.0 Å$^{-1}$ for the light lanthanides are due to multi-electron excitations. **b**, The real component of the Fourier transformed EXAFS data and corresponding fits for the lanthanide complexes, indicating shortening of the average first-shell distance across the Ln series.

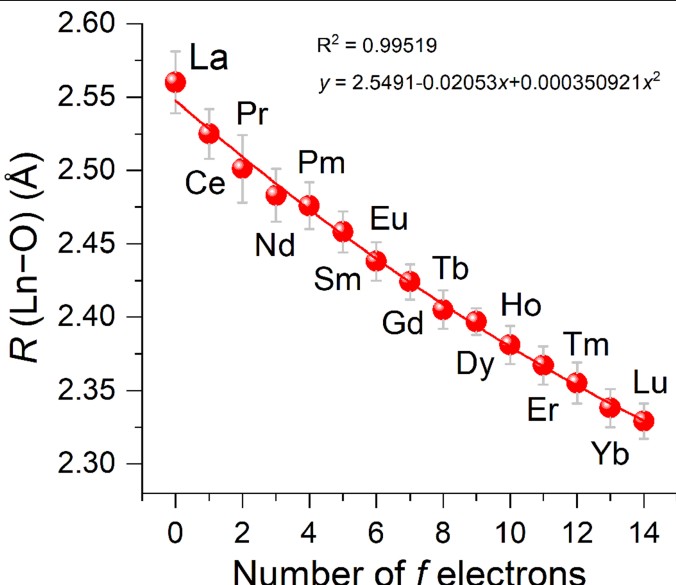

**Extended Data Fig. 7 | Plot of the Ln–O bond distances against the number of 4 $f$ electrons, with the quadratic fit shown as a red line.** The obtained parameters ($b$ = -0.02053 and $c$ = 0.000350921) and a value for $Z_0^*$ = 15.42 (5$p$ electrons) were used to calculate the shielding constant for $f$ electrons ($s$ = 0.74), based on the modified Slater model[36]. 1$\sigma$ error bars in Ln–O bond distance are computed from the covariance matrix of the non-linear minimization of the EXAFS fit[49].

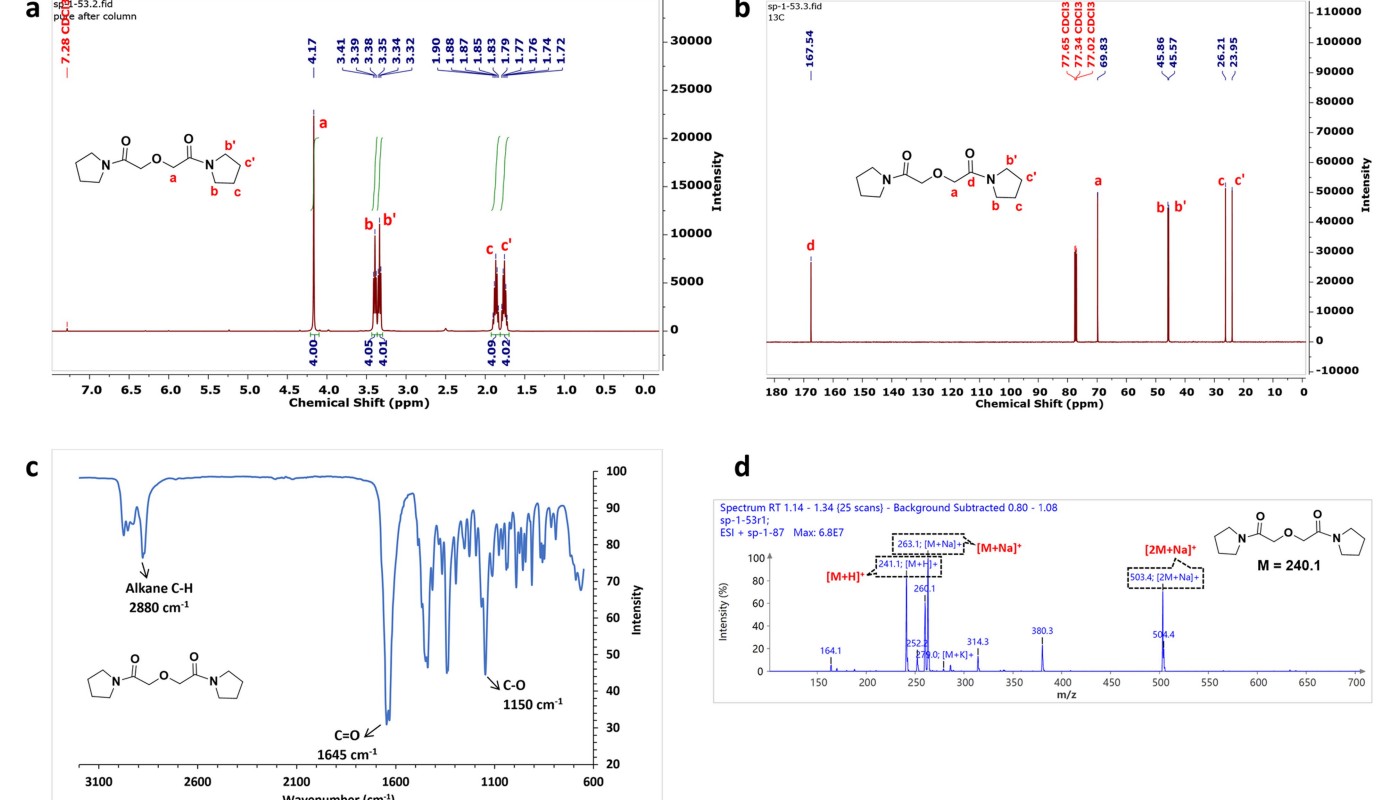

**Extended Data Fig. 8 | PyDGA characterization. a**, [^1]H NMR spectrum of PyDGA in CDCl₃. ¹H NMR (400 MHz, CDCl₃) δ_H 4.17 (s, 4H), 3.39 (t, *J* = 6.9 Hz, 4H), 3.34 (t, *J* = 6.8 Hz, 4H), 1.87 (p, *J* = 6.8 Hz, 4H), 1.76 (p, *J* = 6.6 Hz, 4H). **b**, ¹³C NMR spectrum of PyDGA in CDCl₃. ¹³C NMR (101 MHz, CDCl₃) δ_C 167.54, 69.83, 45.86,

45.57, 26.21, 23.95. **c**, FT-IR spectrum of PyDGA. 2880 (C-H), 1645 (C = O), 1150 (C-O). **d**, ESI-MS ( + Ve) spectrum showing the molecular ion peaks (m/z, Daltons) 241.1 [M + H]⁺; 263.1 [M+Na]⁺; 503.4 [2 M+Na]⁺ of PyDGA using Advion expression compact Mass Spectrometer. Exact mass for [C₁₂H₂₀N₂O₃] was M = 240.1 Daltons.

**Extended Data Table 1 | Summary of EXAFS fitting parameters for [Pm(PyDGA)$_3$]$^{3+}$ including the first and second shells around Pm$^{III}$**

| | |
|---|---|
| CN: Pm–O$^*$; Pm–C$^*$ | 9; 12 |
| 1$^{st}$ shell, $R$ (Å): Pm–O | 2.476 ± 0.016 |
| 2$^{nd}$ shell, $R$ (Å) Pm–C | 3.38 ± 0.07 |
| $\sigma^2_{Pm-O}$ (Å$^2$) | 0.006 ± 0.001 |
| $\sigma^2_{Pm-C}$ (Å$^2$) | 0.02 ± 0.01 |
| L$_3$, $\Delta E_0$ (eV) | 6 ± 2 |
| L$_1$, $\Delta E_0$ (eV) | -1 ± 4 |
| $k$-window: L$_3$; L$_1$ (Å$^{-1}$) | 2.3 – 7.8; 2.4 – 8.0 |
| $R$-window: L$_3$; L$_1$ (Å) | 1.4 – 3.2; 1.3 – 2.45 |

*Footnote:* CN, coordination number assuming amplitude-reduction factor, $S_0^2$=1; $R$ (Å), interatomic distance; $\sigma^2$, Debye-Waller factor; $\Delta E_0$, a single nonstructural parameter for all paths, was varied to align the $k$=0 point of the experimental data and theory. The data were Fourier transformed over a $k$ range of 2.3 Å$^{-1}$ to 7.8 Å$^{-1}$ (L$_3$-edge data) and 2.4 Å$^{-1}$ to 8.0 Å$^{-1}$ (L$_1$-edge data). The data were fitted over an $R$ range of 1.4 Å to 3.2 Å (L$_3$-edge data) and 1.3 Å to 2.45 Å (L$_1$-edge data). The fit contains 6 variables using 10 independent data points (4 degrees of freedom). $^*$Fixed parameters. 1$\sigma$ errors are based on EXAFS fitting uncertainty.

**Extended Data Table 2 | Summary of EXAFS fitting parameters for [Ln(PyDGA)$_3$]$^{3+}$, the entire lanthanide series**

| | CN: Ln–O$^*$; Ln–C$^*$ | 1$^{st}$ shell, $R$ (Å): Ln–O | 2$^{nd}$ shell, $R$ (Å) Ln–C | $\sigma^2_{Ln-O}$ (Å$^2$) | $\sigma^2_{Ln-C}$ (Å$^2$) | L$_3$, $\Delta E_0$ (eV); L$_1$, $\Delta E_0$ (eV) | $k$-window: L$_3$ (Å$^{-1}$) |
|---|---|---|---|---|---|---|---|
| La$^{III}$ | 9; 12 | 2.560(21) | 3.36(6) | 0.010(1) | 0.017(7) | 8(2); -3(3) | 2.3 – 8.5 |
| Ce$^{III}$ | 9; 12 | 2.525(17) | 3.23(5) | 0.010(1) | 0.019(6) | 5(2); -5(3) | 2.3 – 8.8 |
| Pr$^{III}$ | 9; 12 | 2.501(23) | 3.33(10) | 0.009(2) | 0.023(12) | 5(3); -4(4) | 2.3 – 9.2 |
| Nd$^{III}$ | 9; 12 | 2.483(18) | 3.35(9) | 0.008(1) | 0.023(12) | 6(2); -4(3) | 2.3 – 9.7 |
| Pm$^{III}$ | 9; 12 | 2.476(16) | 3.38(7) | 0.006(1) | 0.02(1) | 6(2); -1(4) | 2.3 – 7.8 |
| Sm$^{III}$ | 9; 12 | 2.458(14) | 3.40(6) | 0.007(1) | 0.02(1) | 7(2); -4(3) | 2.3 – 10.2 |
| Eu$^{III}$ | 9; 12 | 2.438(13) | 3.39(6) | 0.009(1) | 0.020(9) | 8(2); -3(3) | 2.3 – 10.9 |
| Gd$^{III}$ | 9; 12 | 2.424(12) | 3.35(5) | 0.0087(9) | 0.019(7) | 9(1); -4(3) | 2.4 – 12.2 |
| Tb$^{III}$ | 9; 12 | 2.405(13) | 3.33(6) | 0.0083(9) | 0.019(8) | 9(1); -5(3) | 2.4 – 12.2 |
| Dy$^{III}$ | 9; 12 | 2.397(9) | 3.34(4) | 0.0085(7) | 0.017(5) | 9(1); -3(3) | 2.4 – 9.6$^{\dagger}$ |
| Ho$^{III}$ | 9; 12 | 2.381(13) | 3.31(5) | 0.0085(9) | 0.018(8) | 9(1); -3(2) | 2.4 – 11.2 |
| Er$^{III}$ | 9; 12 | 2.367(13) | 3.29(6) | 0.0084(9) | 0.018(8) | 9(2); -3(2) | 2.4 – 12.2 |
| Tm$^{III}$ | 9; 12 | 2.355(14) | 3.28(6) | 0.009(1) | 0.018(9) | 8(2); -4(3) | 2.4 – 12.2 |
| Yb$^{III}$ | 9; 12 | 2.338(13) | 3.27(5) | 0.0091(9) | 0.017(7) | 10(2); -2(2) | 2.4 – 12.2 |
| Lu$^{III}$ | 9; 12 | 2.329(12) | 3.27(4) | 0.0093(9) | 0.017(6) | 11(1); -1(3) | 2.4 – 11.5 |

*Footnote:* CN, coordination number assuming amplitude-reduction factor, $S_0^2$=1; $R$ (Å), interatomic distance; $\sigma^2$, Debye-Waller factor; $\Delta E_0$, a single nonstructural parameter for all paths, was varied to align the $k$=0 point of the experimental data and theory. $^*$Fixed parameters. $^{\dagger}$Dy L$_3$ $k$-widow has been restricted due to contamination in the commercial salt source [Dy(NO$_3$)$_3$], containing a small amount of the adjacent lanthanide, Tb$^{III}$. 1$\sigma$ errors are based on EXAFS fitting uncertainty.