## [Peer Review File · Nature]

Manuscript Title: Observation of a promethium complex in solution

Reviewer Comments & Author Rebuttals

Reviewer Reports on the Initial Version:

Referees' comments:

Referee #1 (Remarks to the Author):

This manuscript describes the synthesis and characterisation of a novel promethium compound, with a view to stabilising it for long enough to probe the chemical coordination by X-ray Absorption Spectroscopy. The data detail, for the very first time, the experimentally observed Pm-O bond distance, which is close to previous theoretical predictions. It seems remarkable these days that not all element-O bonding environments are known - that these authors have been able to elucidate this is testament to their approach (and probably a great deal of patience in the radiochemistry laboratory) and the utility of the 6BM beamline at NSLS-II. These results, together with a wider assessment of the entire lanthanide series, are verified by atomistic modelling, which give new, and confirmatory insights to this important series of elements.

The methodology and data produced are sound and well-justified. The authors are to be applauded for their separation approach, which gave rise to such "clean" solutions for analysis. I also understand the rationale behind examining aqueous solutions rather than solid samples; however, do the authors expect there to be any effects from radiolysis of the solutions? Also, I wonder whether the erroneous datapoints in the EXAFS region, at 8 Å⁻¹ in k space, are actually multi-electron excitation peaks rather than the absorption edges of Sm or Nd? The latter is fully justified (although the presence of the edges was not evidenced in the data as far as I could see), but can the authors comment on whether they observed any evidence of the former?

The conclusions are robust and well-justified. I find the references to be appropriate and the abstract to be both enticing and well-written. This is an excellent paper and I fully recommend its publication in Nature, following an answer to the two very minor questions above.

Referee #2 (Remarks to the Author):

This manuscript reports the synthesis and characterisation in solution diffraction measurements of a set of rare-earth complexes of diglycolamide ligands. Importantly, and for the first time, this set of elements includes promethium, the element missing from all previous examples of rare-earth chemistry. This work is therefore highly original and significant, as it provides new knowledge in promethium chemistry

which can be exploited in its separation and its applications, which are burgeoning. Also, its frames promethium in terms of the lanthanide contraction, a feature in the periodic table that has significant consequences, not only in the chemistry of the rare-earth elements but also in that for neighbouring metals such as zirconium and hafnium.

The chemistry presented is very difficult and is only achievable in appropriately equipped laboratories with researchers who are experts. This is the main aspect that makes this work stand out and the authors should be applauded for not only isolating the Pm starting materials and diglycolamide product but also ensuring the validity of the characterisation. As the authors admit, these are not the first Pm complexes to be prepared but they are the first that allows a close inspection of the variation in bond distances for the whole lanthanide series (including La).

There are some weaknesses in the manuscript.

1. The authors state that there is limited comprehension of Pm chemistry. While I agree that there is limited Pm chemistry existing, the general chemistry of this element could be straightforwardly inferred from the related chemistry of its closest congeners, Nd and Sm.

2. I have some misgivings about the EXAFS data analysis as these show only one M-O bond distance that is an average of the 2 x M-O(amide) and 1 x M-O(ether) bonding - the lack of differentiation is ascribed to fluxionality. No evidence is provided for a dynamic process occurring in solution, and while ligand exchange on lanthanides is diffusion controlled, this should not be relevant to complexes of chelating ligands. If exchange is happening (e.g. with water) then this should be taken into account as aquo ligands may be present.

3. The authors describe the subtle changes in Ln-O bond distance across the series (Fig 4a) and state that there are deviations from the standard quadratic model, with a particular deviation prior to Pm. It is interesting that a similar step in the curve is also seen between Tb and Ho, but this is not commented on. Also, if La is removed from the sequence (it is a group 3 element with 0 f electrons) then a more defined line for the series is seen. Can the authors comment on these facets?

Some more general points.

1. I do not like the title - I do not think it reflects the work presented as this is a characterisation of a complex, not just its observation.
2. There appears to be some confusion of what is a rare-earth element and what is a lanthanide. maybe use the term lanthanoid or talk about the lanthanides + La
3. Fig. 1 cation. $\text{Pm}(\text{NO}_3)_3$ is a little too simplistic, more likely to be $[\text{Pm}(\text{H}_2\text{O})_9][\text{NO}_3]_3$.
4. Fig. 2 caption. the complex is not organometallic, but a coordination complex.
5. Fig. 2 caption. It is not clear why the diglycolamide should provide aqueous solubility, especially as the amido oxygen atoms are coordinated.

Referee #3 (Remarks to the Author):

Popovs and co-authors have presented an interesting study of the Promethium complex synthesized with a novel PyDGA complexing agent. The resulting $[\text{Pm}(\text{PyDGA}_3)]^{3+}$ complex was analyzed using experimental X-ray absorption spectroscopy methods, covering XANES and EXAFS regions. The authors have also made great efforts to conduct DFT simulations, shedding light on the nature of observed electronic transitions in the XANES region. Additionally, the EXAFS data was fit with the help of ab initio molecular dynamics simulations. The agreement between theoretical and experimental EXAFS data is excellent. The authors went deeper into their analysis by examining the nature of the Pm-O bond using natural bond orbital calculations. I found the results, particularly the insight that Pm-O bonds originate from an electron density donation from O lone pairs to the Pm center, very exciting.

The manuscript is well-written with flawless English and clear explanations. I believe it will be of interest to the broader Nature community. All the necessary information for understanding the synthesis and characterization of this novel material is provided. Therefore, I am pleased to recommend this paper for publication. However, I have a few remarks and several questions that might be addressed:

- 1) I am somewhat surprised by the Extended data in Fig.2 on Sm, Nd, and Pm compounds. The L3 XANES on different Lns appear very similar. Our experience in investigating a series of Ln ions in the same structure typically shows differences between them, and this is usually reflected in the XANES data, specifically in the positions of the post-edge features. (c.f. Zasimov et al, Inorg.Chem. 2022)

However, I don't observe this in the $[\text{Ln}(\text{PyDGA}_3)]^{3+}$ particular case. Nevertheless, the EXAFS data does show such trends (Fig. 4). I'm curious as to why the position of post-edge features in XANES does not appear sensitive to these differences. It might be helpful if the authors could add dashed vertical lines to indicate the trends in post-edge features. Do the authors have an explanation for this?

- 2) I found Extended Figure 4 a bit misleading. The authors mention that the figure should show region "I" corresponding to the 2p-5d transitions and region "III" attributed to transitions involving Pm 4f orbitals, while the origin of broad feature "IV" is quite complex with leading components from 2p to 5d/ligand and Pm 4f dz3 orbitals (Extended Data Fig. 4). However, only the plot of molecular orbitals is provided, which might not be understandable to non-expert readers. I suggest plotting the corresponding density of states related to the 5d and 4f contributions below the spectrum.
- 3) I'm curious if the pre-edge region of the Pm compound only shows 5d states reflections (please see my comment above; it would be better to examine the DOS with respect to the Fermi energy). It is somehow known fact that region "I" in Ln^{3+} should contain pre-edge structure due to the quadrupole 2p-4f excitations (which might only be visible in high-energy resolution XAS mode like HERFD, as authors may not be able to experimentally resolve it). However, DFT calculations should show the 4f contribution to Region I in the spectrum. Can the authors comment on why they don't observe it? (please see more work done at the pre-edges : Hämäläinen, K., Siddons, D. P., Hastings, J. B. & Berman, L. E. Elimination of the inner-shell lifetime broadening in x-ray-absorption spectroscopy. *Phys. Rev. Lett.* **67**, 2850–2853 (1991). Kvashnina, K. O., Butorin, S. M.

& Glatzel, P. Direct study of the f-electron configuration in lanthanide systems. *J. Anal. At. Spectrom.* **26**, 1265 (2011), Zaslavov, P. *et al.* HERFD-XANES and RIXS Study on the Electronic Structure of Trivalent Lanthanides across a Series of Isostructural Compounds. *Inorg. Chem.* **61**, 1817–1830 (2022). and citations there)

Out of curiosity, I took the Pm₂O₃ structure from 1972 and ran FEFF calculations to get an idea of the 4f and 5d states distributions. I believe your Pm L3 spectrum might contain 4f states in Region I.

- 4) I recommend including information about the ground state configuration. How many 5d and 4f electrons does Pm have? I ask because I am a bit confused. Figure 4c indicates that the Pm³⁺ compound contains 4f¹ electrons. However, the known ground state configuration of Pm is [Xe] 4f⁵ 6s² with the term symbol 6H_{5/2}. Therefore, Pm³⁺ should contain 4 electrons at the 4f level. Do authors see it differently?

Minor comments:

- What was the Pm activity per sample? (perhaps in Bq)
- What was the final Pm concentration in the sample measured by XAS?
- I understand that the sample was double confined, but with which material (I see polyamide mentioned later, but what was the thickness)? How did the sample holder look? Perhaps authors can add a photo of it to the Extended Data file? (I only saw photo on how the synthesis was done) How was it transported to the beamline from the lab?
- Samples were measured at room temperature. Did the authors observe any radiation damage, or has this issue been checked?
- How long was the EXAFS data collected (few min or 30 minutes per scan)? Was it taken from one spot on the sample or from several spots? How homogeneous was the sample?
- What was the beam size?
- Line 259 in the Data Collection section: It is written that "the data were energy-calibrated to the main edge from the spectra of Ln oxide standards," but earlier in the text, authors mention that energy calibration was done using the Fe foil. Please add more clarity.
- Beamline 6-BM of NSLS was used, but no citation to the BL is given. Please add it.
- Extended Figure 8 is identical to Figure 4c. I believe the Extended Data can be removed.

Author Rebuttals to Initial Comments:

Referee #1 (Remarks to the Author):

This manuscript describes the synthesis and characterisation of a novel promethium compound, with a view to stabilising it for long enough to probe the chemical coordination by X-ray Absorption Spectroscopy. The data detail, for the very first time, the experimentally observed Pm-O bond distance, which is close to previous theoretical predictions. It seems remarkable these days that not all element-O bonding environments are known - that these authors have been able to elucidate this is testament to their approach (and probably a great deal of patience in the radiochemistry laboratory) and the utility of the 6BM beamline at NSLS-II. These results, together with a wider assessment of the entire lanthanide series, are verified by atomistic modelling, which give new, and confirmatory insights to this important series of elements.

We appreciate the referee's critical review of our manuscript.

The methodology and data produced are sound and well-justified. The authors are to be applauded for their separation approach, which gave rise to such "clean" solutions for analysis. I also understand the rationale behind examining aqueous solutions rather than solid samples; however, do the authors expect there to be any effects from radiolysis of the solutions?

We thank the referee for the valid comments. No radiation damage was observed during our XAS measurements, as we utilized a low-flux-density, unfocused beam at 6-BM. This was verified by comparing individual XAS scans for all Ln samples, which did not show any abnormal changes. Additionally, a comparison of individual Pm-PyDGA XANES scans from the L₃- and L₁-edge spectra is also provided in Extended Data Fig. 13.

Also, I wonder whether the erroneous datapoints in the EXAFS region, at 8 Å⁻¹ in k space, are actually multi-electron excitation peaks rather than the absorption edges of Sm or Nd? The latter is fully justified (although the presence of the edges was not evidenced in the data as far as I could see), but can the authors comment on whether they observed any evidence of the former? We have attached the Pm XAFS plot with the magnified region (inset), pointing to the small presence of Sm L₃ (6716 eV)/Nd L₂ (6722 eV) edges.

In Fig. 3b (main text), the XAS data are presented in k -space, and the $\chi(k)$ is amplified by k^3 , making the adjacent lanthanides' features more apparent (quick calculations show that $\sim 8.2 \text{ \AA}^{-1}$ is in line with the energy range for the Sm L_3 /Nd L_2 edges). We also note in the Methods section that “a small quantity of ^{146}Nd was present in the sample due to challenging separations of the adjacent lanthanides using the aforementioned techniques. Traces of Sm present in the sample on the moment of XAS measurements originated from the radioactive decay of promethium to the daughter samarium according to the following process: $^{147}\text{Pm} (\beta^- \rightarrow) ^{147}\text{Sm}$. Approximately 77 days had passed between the Pm purification and XAS data collection. Based on the $\tau_{1/2} = 2.62$ years of the radioactive decay, up to 5.632% of the starting ^{147}Pm has decayed into ^{147}Sm at the time of the sample measurements at NSLS-II.”

Regarding the multi-electron excitation peaks mentioned by the referee, we indeed observe them at approximately 5.7 \AA^{-1} to 6.0 \AA^{-1} for the light lanthanides (Extended Data Fig. 7). This is consistent with the previous study by Solera et al. (Multielectron Excitations at the L Edges in Rare Earth Ionic Aqueous Solutions. Phys. Rev. B 1995, 51, 2678–2686), where the multi-electron excitation features were seen at about $5\text{-}7 \text{ \AA}^{-1}$.

The conclusions are robust and well-justified. I find the references to be appropriate and the abstract to be both enticing and well-written. This is an excellent paper and I fully recommend its publication in Nature, following an answer to the two very minor questions above.

We appreciate the referee's positive evaluation of our work.

Referee #2 (Remarks to the Author):

This manuscript reports the synthesis and characterisation in solution diffraction measurements of a set of rare-earth complexes of diglycolamide ligands. Importantly, and for the first time, this set of elements includes promethium, the element missing from all previous examples of rare-earth chemistry. This work is therefore highly original and significant, as it provides new knowledge in promethium chemistry which

can be exploited in its separation and its applications, which are burgeoning. Also, it frames promethium in terms of the lanthanide contraction, a feature in the periodic table that has significant consequences, not only in the chemistry of the rare-earth elements but also in that for neighbouring metals such as zirconium and hafnium.

We appreciate the referee's comments.

The chemistry presented is very difficult and is only achievable in appropriately equipped laboratories with researchers who are experts. This is the main aspect that makes this work stand out and the authors should be applauded for not only isolating the Pm starting materials and diglycolamide product but also ensuring the validity of the characterisation. As the authors admit, these are not the first Pm complexes to be prepared but they are the first that allows a close inspection of the variation in bond distances for the whole lanthanide series (including La).

We appreciate the referee's positive evaluation of our work.

There are some weaknesses in the manuscript.

1. The authors state that there is limited comprehension of Pm chemistry. While I agree that there is limited Pm chemistry existing, the general chemistry of this element could be straightforwardly inferred from the related chemistry of its closest congeners, Nd and Sm.

We partially agree with the comment from the reviewer. We note that Pm is situated between Nd, which has accessible oxidation states of +3 and +4, and Sm, which exhibits oxidation states of +2 and +3. By stating that Pm chemistry had been "virtually unexplored", we tried to draw the community's attention to the fact that this element is indeed squarely located in the lanthanide series, however very little is known about its chemical behavior. In fact, it is plausible that Pm could exhibit chemical properties, such as an accessible oxidation state, that are different from one or, perhaps, even both of its neighbors and further studies are warranted to explore this possibility. For instance, it is not known whether complexes with formal Pm(II) or Pm(IV) can be prepared.

2. I have some misgivings about the EXAFS data analysis as these show only one M-O bond distance that is an average of the 2 x M-O(amide) and 1 x M-O(ether) bonding - the lack of differentiation is ascribed to fluxionality. No evidence is provided for a dynamic process occurring in solution, and while ligand exchange on lanthanides is diffusion controlled, this should not be relevant to complexes of chelating ligands. If exchange is happening (e.g. with water) then this should be taken into account as aquo ligands may be present.

We appreciate the reviewer's comment and would like to provide clarification regarding our use of the term 'dynamic process' concerning the behavior of ligands around the metal ion in solution. The interactions governing ion coordination in an aqueous solution differ significantly from those in a crystalline solid. In solid-state environments, structural arrangements are predominantly static, allowing for distinct resolution of Ln-O (amide) and Ln-O (ether) from corresponding single-crystal XRD data, often collected for solid-state complexes at ~100 K. However, in a solution, molecules and ions lack organization on a lattice, causing the bonds formed between the metal ion and the donor groups of the ligand to exhibit dynamic behavior.

Our EXAFS data for the Pm complex were obtained in the solution phase at room temperature. Under these conditions, the primarily ionic bonding interactions between Pm(III) and the donor atoms of PyDGA are expected to be more dynamic compared to the solid-state. This dynamic behavior contributes to the challenge of resolving individual Pm–O (amide) and Pm–O (ether) distances in the complex. Our fitting model, describing the Pm EXAFS with an average Pm–O distance, proved satisfactory, and no statistically significant value could be added to the EXAFS analysis by considering separate Pm–O (amide) and Pm–O (ether) distances.

Moreover, the dynamic nature of the Pm(III)-ligand interactions is visually demonstrated in the attached movie (mp4 file), illustrating the Pm coordination complex and water molecules within 5 Å of the metal ion. This visualization is based on our *ab-initio* molecular dynamics simulations, which also reproduce the experimental EXAFS spectra (Fig. 3b,c). Additionally, we have included a plot showing the evolution and overlap of the Pm–O (amide) and Pm–O (ether) distances over time. As depicted, it is statistically challenging to distinguish individual Pm–O (amide) and Pm–O (ether) bonds in solution. Therefore, we attribute the lack of differentiation to the fluxionality of these interactions in an aqueous environment at room temperature.

3. The authors describe the subtle changes in Ln-O bond distance across the series (Fig 4a) and state that there are deviations from the standard quadratic model, with a particular deviation prior to Pm. It is interesting that a similar step in the curve is also seen between Tb and Ho, but this is not commented on. Also, if La is removed from the sequence (it is a group 3 element with 0 f electrons) then a more defined line for the series is seen. Can the authors comment on these facets?

We appreciate the insightful comment from the reviewer. The EXAFS data in Fig. 4c are presented with error bars, specifically 1σ error bars associated with each data point. These are based on EXAFS fitting uncertainty and were computed from the covariance matrix of the non-linear minimization of the EXAFS fit. Therefore, drawing definitive conclusions from a small deviation from the standard quadratic model becomes challenging, especially considering potential influences such as the restricted k -window for the Pm and Dy cases (see Extended Data Table 4).

In the manuscript, we just state that there is a “somewhat accelerated shortening of bonds at the beginning of lanthanide series” and “filling the 4f orbitals apparently influences shielding of the nuclear charge and according to our data this effect was most pronounced early in the series from La to Pm, accounting for as much as $\approx 36\%$ of the overall Ln contraction.” These findings align with Shannon’s effective ionic radii decrease (at a coordination number of nine), which is more substantial at the beginning of the series than at the end.

It is important to note that our use of a quadratic fit was aimed at obtaining parameters necessary for the derivation of a shielding constant for f electrons ($s = 0.74$). While we agree that removing La from the sequence could result in a more defined line, including La was crucial for comparing the f-electron shielding constants obtained by different methods. To the best of our knowledge, a generally accepted value of $s = 0.69$ was obtained from the Ln ionization energies, including La in the dataset.

Some more general points.

1. I do not like the title - I do not think it reflects the work presented as this is a characterisation of a complex, not just its observation.

We thank the reviewer for this suggestion. Merriam-Webster dictionary provides the following definition of the noun “observation”—*an act of recognizing and noting a fact or occurrence often involving measurement with instruments*. We believe the title corresponds to the spirit of our work.

2. There appears to be some confusion of what is a rare-earth element and what is a lanthanide. maybe use the term lanthanoid or talk about the lanthanides + La.

We appreciate the reviewer’s attention to details and the opportunity to clarify the terminology used in our manuscript. In the context of our work, we have chosen to use the term “lanthanide” to refer to the series of chemical elements with atomic numbers 57 to 71, inclusive. We have included the following text in the introduction paragraph to elucidate our use of the term “lanthanide”: “One reason promethium (Pm) was so elusive for many years, despite a relatively low atomic number (Fig. 1), is that it is the only element in the lanthanide series (elements with atomic numbers 57–71) with no stable isotopes.”

3. Fig. 1 cation. $\text{Pm}(\text{NO}_3)_3$ is a little too simplistic, more likely to be $[\text{Pm}(\text{H}_2\text{O})_9][\text{NO}_3]_3$.

We thank the reviewer for this correction. The promethium nitrate aqueous solution was slowly evaporated in a negative-pressure radiological glovebox upon heating to obtain an anhydrous salt. The photograph was taken right after the sample of an anhydrous salt cooled to reach ambient temperature. There were no additional precautions taken to ensure the complete removal of water from the glovebox. Additionally, knowing that anhydrous $\text{Pm}(\text{NO}_3)_3$, similar to other lanthanides, is expected to be hygroscopic, we cannot rule out that a small amount of water was reabsorbed by the salt to form a hydrate. Therefore, per reviewer’s suggestion we have changed figure caption to reflect this fact: “**Fig. 1 | Summary of lanthanide elements, year of their discovery, and electron configuration after [Xe] core along with a photograph of purified Pm^{III} compound prepared in this study.** The depicted pink-colored

$^{147}\text{Pm}(\text{NO}_3)_3 \cdot n\text{H}_2\text{O}$ ($n < 9$) solid residue was obtained after multiple purification steps and used in a Pm^{III} complexation with the diglycolamide ligand.”

4. Fig. 2 caption. the complex is not organometallic, but a coordination complex.

We have changed Fig. 2 caption to coordination complex: “**Fig. 2 | The synthesized multidentate ligand bipyrrolidine diglycolamide (PyDGA) chelates Pm^{III} to form the homoleptic coordination complex in an aqueous solution.**”

5. Fig. 2 caption. It is not clear why the diglycolamide should provide aqueous solubility, especially as the amido oxygen atoms are coordinated.

We thank the reviewer for the comment. PyDGA ligand was designed and synthesized to serve several purposes. First, the ligand’s substitution pattern was specifically ensuring good aqueous solubility while maintaining strong binding and similar functionality as in a very promising reagent used in lanthanide separation, TODGA (N,N,N',N'-tetraoctyl diglycolamide) that is extremely lipophilic. The present PyDGA ligand has experimentally verified aqueous solubility in excess of 200 g/L. Second, we used a neutral chelating ligand PYDGA to ensure high aqueous solubility of the complex formed with the lanthanides to prevent the possible precipitation of the metal complex.

Referee #3 (Remarks to the Author):

Popovs and co-authors have presented an interesting study of the Promethium complex synthesized with a novel PyDGA complexing agent. The resulting $[\text{Pm}(\text{PyDGA}_3)]^{3+}$ complex was analyzed using experimental X-ray absorption spectroscopy methods, covering XANES and EXAFS regions. The authors have also made great efforts to conduct DFT simulations, shedding light on the nature of observed electronic transitions in the XANES region. Additionally, the EXAFS data was fit with the help of ab initio molecular dynamics simulations. The agreement between theoretical and experimental EXAFS data is excellent. The authors went deeper into their analysis by examining the nature of the Pm-O bond using natural bond orbital calculations. I found the results, particularly the insight that Pm-O bonds originate from an electron density donation from O lone pairs to the Pm center, very exciting. The manuscript is well-written with flawless English and clear explanations. I believe it will be of interest to the broader Nature community. All the necessary information for understanding the synthesis and characterization of this novel material is provided. Therefore, I am pleased to recommend this paper for publication.

We thank the referee for thoughtful review and positive feedback on our work. The insightful comments have significantly contributed to the improvement of the manuscript.

However, I have a few remarks and several questions that might be addressed:

1. I am somewhat surprised by the Extended data in Fig.2 on Sm, Nd, and Pm compounds. The L3 XANES on different Ln appear very similar. Our experience in investigating a series of Ln ions in the same

structure typically shows differences between them, and this is usually reflected in the XANES data, specifically in the positions of the post-edge features. (c.f. Zaslavov et al, *Inorg.Chem.* 2022). However, I don't observe this in the Ln(PyDGA3)]₃⁺ particular case. Nevertheless, the EXAFS data does show such trends (Fig. 4). I'm curious as to why the position of post-edge features in XANES does not appear sensitive to these differences. It might be helpful if the authors could add dashed vertical lines to indicate the trends in post-edge features. Do the authors have an explanation for this?

We indeed observe changes in the positions of the post-edge features, which was not much evident from Extended Data Fig. 3, because of the plotting scale. We have introduced data for the La and Lu complexes and added vertical lines to make the shift in the post-edge positions more apparent. The trends in post-edge XANES features along the lanthanide series agreed well with the previous study by Zaslavov et al. (cited in the revised version of the manuscript). The following sentence was added: "Furthermore, the shrinkage of the Ln–O bonds is corroborated by the trend in the relative energy positions of the Ln L₃-edge XANES spectral features (Extended Data Fig. 3), consistent with the results of a recent study⁴¹ on some isostructural Ln compounds using high-energy-resolution fluorescence-detected XANES (HERFD-XANES)^{42,43} measurements." Extended Data Fig. 3 was also updated accordingly.

2. I found Extended Figure 4 a bit misleading. The authors mention that the figure should show region "I" corresponding to the 2p-5d transitions and region "III" attributed to transitions involving Pm 4f orbitals, while the origin of broad feature "IV" is quite complex with leading components from 2p to 5d/ligand and Pm 4f d_{z²} orbitals (Extended Data Fig. 4). However, only the plot of molecular orbitals is provided, which might not be understandable to non-expert readers. I suggest plotting the corresponding density of states related to the 5d and 4f contributions below the spectrum.

Since we focus on an isolated molecular system (the Pm complex), the concept of density of states (DOS), primarily used in the solid-state physics to represent the number of states in unit energy interval with energy levels being contiguous, is not very relevant here because in a molecular system the energy levels are discrete. Therefore, we followed the original DFT/ROCIS approach and XANES interpretation by Dr. F. Neese (the DFT/ROCIS code developer), where the assignment of XANES features is interpreted on the basis of natural difference orbitals (NDOs).

Additionally, following the reviewer's recommendation, we performed multiple scattering theory calculations (FEFF9 code) to reproduce XANES and explore the origin of the peaks using DOS. The corresponding FEFF9-simulated XANES spectrum and the local DOS are plotted in Extended Data Fig. 4, consistent with our DFT/ROCIS results. We anticipate that presenting results in this way (both molecular orbitals and DOS are presented) will be helpful to a general reader.

3. I'm curious if the pre-edge region of the Pm compound only shows 5d states reflections (please see my comment above; it would be better to examine the DOS with respect to the Fermi energy). It is somehow known fact that region "I" in Ln³⁺ should contain pre-edge structure due to the quadrupole 2p-4f excitations (which might only be visible in high-energy resolution XAS mode like HERFD, as authors may not be able to experimentally resolve it). However, DFT calculations should show the 4f contribution to Region I in the spectrum. Can the authors comment on why they don't observe it? (please see more work done at the pre-edges : Hämmäläinen, K., Siddons, D. P., Hastings, J. B. & Berman, L. E. Elimination of the inner-shell lifetime broadening in x-ray-absorption spectroscopy. *Phys. Rev. Lett.* 67, 2850–2853

(1991). Kvashnina, K. O., Butorin, S. M. & Glatzel, P. Direct study of the f-electron configuration in lanthanide systems. *J. Anal. At. Spectrom.* 26, 1265 (2011), Zsimev, P. et al. HERFD-XANES and RIXS Study on the Electronic Structure of Trivalent Lanthanides across a Series of Isostructural Compounds. *Inorg. Chem.* 61, 1817–1830 (2022). and citations there)

Out of curiosity, I took the Pm₂O₃ structure from 1972 and ran FEFF calculations to get an idea of the 4f and 5d states distributions (please see attached file). I believe your Pm L3 spectrum might contain 4f states in Region I.

Since the conventional L₃-edge XANES of Pm is likely broadened by the large 2p core-hole lifetime, we acknowledge that high-energy-resolution fluorescence-detected XANES (HERFD-XANES) measurements could potentially offer better resolution of XANES features for Pm and other lanthanides. However, achieving this would necessitate access to specific beamline/instrumentation. Additionally, the challenges associated with handling the Pm radionuclide would warrant a separate publication on this topic. We thank the reviewer for providing relevant references on HERFD-XANES works for the Ln, which we cite in the revised version of our manuscript.

We also appreciate the reviewer for supplying additional FEFF calculations for the Pm₂O₃ structure. These calculations demonstrate that the pre-edge feature originates from both f and d states, as seen in the projected DOS. Our FEFF9 calculations for the Pm(PyDGA)₃ complex (Extended Data Fig. 4b) led to similar conclusions, indicating that the pre-edge feature (I) in the Pm complex arises from quadrupole 2p-4f and dipole 2p-5d electronic transitions.

A detailed analysis of our DFT/ROCIS results using the natural difference orbitals (NDOs) revealed that transitions from 2p to 5d orbitals mainly contribute to the pre-edge region (I), with a smaller fraction of 2p to 4f transitions. This is generally consistent with the FEFF9-DOS calculations. While quadrupole transitions (2p-4f) are typically two orders of magnitude weaker than dipole transitions (2p-5d), the DFT/ROCIS theory might introduce variations in relative values of 4f and 5d transition energies, depending on the chosen density functional and/or basis set. Thus, although estimating the exact contribution of 4f and 5d transitions in region I is challenging, the consistent results from the molecular orbital approach (DFT/ROCIS) and DOS (multiple scattering theory) indicate the involvement of both f and d states in region I.

We have modified the following sentence in the main text: “Based on our density functional theory restricted open shell configuration interaction singles (DFT/ROCIS) and multiple scattering theory calculations (Extended Data Fig. 4), region “I” corresponds to transitions from Pm 2p to 4f/5d orbitals and the most intense peak “II” is dominated by 2p core electron excitations to 5d but with some PyDGA orbital contributions. Less visible peak “III” can be attributed to transitions involving Pm 4f/5d/ligand orbitals, while the origin of broad feature “IV” is quite complex with leading components from 2p to 5d/ligand and Pm 4f dz³ orbitals.” Extended Data Fig. 4 was also updated accordingly.

4. I recommend including information about the ground state configuration. How many 5d and 4f electrons does Pm have? I ask because I am a bit confused. Figure 4c indicates that the Pm³⁺ compound contains 4f¹ electrons. However, the known ground state configuration of Pm is [Xe] 4f⁵ 6s² with the term symbol 6H_{5/2}. Therefore, Pm³⁺ should contain 4 electrons at the 4f level. Do authors see it differently?

We confirm that the Pm(III) complex contains 4f4 electrons (not 4f1). Figure 4c in the original manuscript was also correct. X-axis in Figure 4c corresponds to the number of Ln f-electrons in the studied complexes; i.e. 4f1 corresponds to Ce(III) and 4f4 to Pm(III). Computational details Methods section also indicates the ground state configuration of the Pm(III) complex: "... the complex was treated as a triply charged quintet with four unpaired *f*-electrons."

Minor comments:

- What was the Pm activity per sample? (perhaps in Bq)

For 8.5 mM ¹⁴⁷Pm, the activity was 4.0 GBq (108 mCi).

- What was the final Pm concentration in the sample measured by XAS?

This information is provided in the Methods section: "Approximately 77 days had passed between the Pm purification and XAS data collection. Based on the $\tau_{1/2} = 2.62$ years of the radioactive decay, up to 5.632% of the starting ¹⁴⁷Pm has decayed into ¹⁴⁷Sm at the time of the sample measurements at NSLS-II." Initially, 8.5 mM Pm was used for the sample preparation right after the purification procedure, and thus the final Pm concentration measured by XAS was ~8 mM.

- I understand that the sample was double confined, but with which material (I see polyamide mentioned later, but what was the thickness)? How did the sample holder look? Perhaps authors can add a photo of it to the Extended Data file? (I only saw photo on how the synthesis was done) How was it transported to the beamline from the lab?

This information has been provided in the Methods section. We also added the capillary thickness and manufacturer (0.05 mm, Cole-Parmer): "To ensure the full complexation of promethium, a solution (~90 μ L) of 8.5 mM ¹⁴⁷Pm(NO₃)₃ was added to 180 mM PyDGA. The obtained solution was then loaded into a polyimide capillary (1.8 mm inner diameter by 5 cm long, 0.05 mm thickness, Cole-Parmer) using a Hamilton syringe and then sealed twice with Devcon 2 Ton epoxy (Extended Data Fig. 1). Once the epoxy had dried completely, the sample was transferred from a glovebox to a radiological fume hood for further decontamination. The sample was then surveyed and doubly contained for shipment to the XAS beamline."

We have provided additional photos related to the sample transportation and our sample holder in Extended Data Fig.1.

- Samples were measured at room temperature. Did the authors observe any radiation damage, or has this issue been checked?

We did not observe any radiation damage during our XAS measurements, as the low-flux-density, unfocused beam was used at 6-BM. This was checked by comparing individual XAS scans for all Ln samples, which did not show any abnormal changes. Additionally, a comparison of individual Pm-PyDGA XANES scans from the L₃- and L₁-edge spectra is also provided in Extended Data Fig. 13.

- How long was the EXAFS data collected (few min or 30 minutes per scan)? Was it taken from one spot on the sample or from several spots? How homogeneous was the sample?

We spent approximately 10 minutes per scan, focusing the beam on one spot. As we only had liquid samples, homogeneity was not an issue.

- What was the beam size?

6 mm x 0.3 mm

- Line 259 in the Data Collection section: It is written that "the data were energy-calibrated to the main edge from the spectra of Ln oxide standards," but earlier in the text, authors mention that energy calibration was done using the Fe foil. Please add more clarity.

This statement is correct – calibration for the Pm sample was done using an Fe foil. For other lanthanides, we used the corresponding Ln oxide standards. This is now clarified in the text: "The data for Ln, except Pm, were energy-calibrated to the main edge from the spectra of Ln oxide standards."

- Beamline 6-BM of NSLS was used, but no citation to the BL is given. Please add it.

There isn't one yet. The beamline was fully acknowledged in the acknowledgment section.

- Extended Figure 8 is identical to Figure 4c. I believe the Extended Data can be removed.

We used Extended Data Fig. 8 to illustrate the quadratic fit and the derived parameters from this fit, which were then employed to calculate the shielding constant for f electrons. The figure caption has been slightly modified to accurately reflect this.

Reviewer Reports on the First Revision:

Referees' comments:

Referee #2 (Remarks to the Author):

I very much appreciate the time and effort that the authors have spent in explaining aspects of this manuscript and providing more information. The majority of my concerns are alleviated and the changes and additions to the manuscript are welcome.

I have an issue that remains concerning the averaging of the Pm-O bond distances and the dynamic processes occurring in solution. The movie of the dynamic process is very helpful and it is interesting to see that the diglycolamide ligand is locked in position by the ether-O coordination with the amide-O atoms undergoing association-dissociation. Can the authors confirm that they have modelled a situation in which a molecule of water is bound to Pm from the start and not outer-sphere as shown in the movie, i.e., $[\text{Pm}(\text{OH}_2)(\text{L})_3][(\text{OH}_2)_2]$? It would be important to see if this water molecule is immediately displaced from the Pm cation by the pendant amide-O of a diglycolamide ligand as, if this is not the case, then the EXAFS data would represent averaging of the diglycolamides and water molecules at Pm.

The authors provide some solid-state comparison in which Sm and Eu glycolamides are charge-balanced by nitratometalates of these elements. Can the authors comment on the anionic part of the Pm complex? What is present in solution, simple NO_3^- anions or a more complex $[\text{Pm}(\text{NO}_3)_6]^{3-}$ metalate? If the former case, is it assumed that the nitrates interact with the complex cation in a similar manner to that seen by Shafer to form trefoil knot-like structures?

Referee #3 (Remarks to the Author):

The authors have addressed all the issues I raised. I do not have any further comments. Thank you! Great manuscript!

Author Rebuttals to First Revision:

Referee #2 (Remarks to the Author) 2-nd round of reviews:

I very much appreciate the time and effort that the authors have spent in explaining aspects of this manuscript and providing more information. The majority of my concerns are alleviated and the changes and additions to the manuscript are welcome.

I have an issue that remains concerning the averaging of the Pm-O bond distances and the dynamic processes occurring in solution. The movie of the dynamic process is very helpful and it is interesting to see that the diglycolamide ligand is locked in position by the ether-O coordination with the amide-O atoms undergoing association-dissociation.

We thank the referee for the additional comments. We would like to clarify that the Pm-O bonds in the movie do not undergo association-dissociation. Instead, based on our molecular dynamics simulations, the movie illustrates that Pm-O bonds in solution are dynamic, with the atoms oscillating around an equilibrium distance.

Can the authors confirm that they have modelled a situation in which a molecule of water is bound to Pm from the start and not outer-sphere as shown in the movie, i.e., $[\text{Pm}(\text{OH}_2)(\text{L})_3][(\text{OH}_2)_2]$? It would be important to see if this water molecule is immediately displaced from the Pm cation by the pendant amide-O of a diglycolamide ligand as, if this is not the case, then the EXAFS data would represent averaging of the diglycolamides and water molecules at Pm.

We utilized *ab initio* molecular dynamics simulations in our study. Generating a trajectory long enough to observe the displacement of water from the metal center would be computationally prohibitive.

Our EXAFS fits, which combine components from the three PyDGA ligands and additional water molecule(s) in the first coordination sphere, were found to not accurately describe the data set. This suggests that water molecule(s) are unlikely to be present in the inner shell at the experimental conditions. Furthermore, considering the metal ion-to-ligand ratio of approximately 1:20 and high stability constants ($\log \beta_3$) exhibited by the DGA family of ligands (*J. Phys. Chem. B* 2017, 121, 12, 2640), it is expected that the homoleptic 1:3 species will dominate under the conditions of our experiment.

The authors provide some solid-state comparison in which Sm and Eu glycolamides are charge-balanced by nitratometalates of these elements. Can the authors comment on the anionic part of the Pm complex? What is present in solution, simple NO_3^- anions or a more complex $[\text{Pm}(\text{NO}_3)_6]^{3-}$ metalate?

A dilute aqueous solution of ^{147}Pm (90 μL , 8.5 mM) was studied. Therefore, NO_3^- anions are present in the solution, as the metal cations are expected to be complexed by PyDGAs, given the $\approx 1:20$ metal ion-to-ligand ratio and high stability constants.

If the former case, is it assumed that the nitrates interact with the complex cation in a similar manner to that seen by Shafer to form trefoil knot-like structures?

Assuming that we are referring to the following article (Baldwin, A. G., Ivanov, A. S., Williams, N. J., Ellis, R. J., Moyer, B. A., Bryantsev, V. S., & Shafer, J. C. (2018). Outer-sphere water clusters tune the lanthanide selectivity of diglycolamides. *ACS central science*, 4(6), 739-747), in which one of us is a co-author, the

nitrates are screened by water molecules away from the $\text{Ln}(\text{DGA})_3^{3+}$ complex in an aqueous environment. The formation of trefoil knot-like structures, where nitrates are closely associated with the Ln complexes, is expected in a non-polar, low dielectric constant medium (e.g., organic phase).